# A model of $\beta$-adrenergic stimulation in human ventricular cells for tissue-scale simulations of sympathetically modulated tachycardias

Kelly Zhang[1] , Karl Magtibay[2,3] , Natalia Trayanova[1] and Edward Vigmond[2,3]

[1]*Department of Biomedical Engineering, Johns Hopkins University, Baltimore, Maryland, USA*
[2]*Institut de Mathématiques de Bordeaux, Université de Bordeaux, Talence, Gironde, France*
[3]*IHU L'institut des maladies du rythme cardiaque (Liryc), Hôpital Xavier Arnozan, Pessac, Gironde, France*

Handling Editors: Bjorn Knollmann & Eleonora Grandi

The peer review history is available in the Supporting Information section of this article (https://doi.org/10.1113/JP289340#support-information-section).

**Abstract figure legend** Modelling $\beta$-adrenergic stimulation in sympathetically remodelled ventricular substrates. Myocardial infarction could result in sympathetic remodelling in ventricular tissues, which affects arrhythmia dynamics. However sympathetic remodelling is challenging to control experimentally; therefore we created an ionic model that

**Kelly Zhang** is a PhD candidate in the Department of Biomedical Engineering at Johns Hopkins University. She earned her B.S. in bioengineering with a minor in mathematics from the University of Washington in 2022. Her current research focuses on developing multiscale computational models to study how autonomic nervous system dysregulation contributes to the initiation and dynamics of ventricular tachycardia. Kelly's broader goal is to translate computational modelling advances into tools that guide patient-specific therapies and clinical decision-making in cardiac electrophysiology. **Karl Magtibay** obtained his degrees (BEng '12, MASc. '14, and PhD '25) in biomedical engineering from Toronto Metropolitan University (formerly Ryerson University), Canada. At the time of writing, Karl is a postdoctoral researcher at the Université de Bordeaux and a member of the cardiac modelling team at

the IHU L'institut des maladies du rythme cardiaque (Liryc). Karl's research is focused on modelling autonomic effects to cardiac electrophysiology for developing tools and methods for managing arrhythmias. Karl's research interests include signal and image processing and analysis, device development and simulations with applications to cardiac electrophysiology.

K. Zhang and K. Magtibay share first authorship.

incorporates the macroscopic effects of $\beta$-adrenergic stimulation and could be implemented in tissue-level simulations. We showed that our model could replicate the physiological effects of $\beta$-adrenergic stimulation by reducing action potential duration. Moreover we showed that higher densities of sympathetic remodelling in ventricular substrates could host faster and more organized rotors than those hosted in a non-sympathetically stimulated, homogeneous substrate with a critical spatial density of $\geq 15\%$. Finally spatial gradients of sympathetic remodelling splinter a rotor to several slow and disorganized wavelets with a critical spatial gradient of $\geq \Delta 15\%$.

**Abstract** Ventricular tachycardia (VT) dynamics in myocardial infarction and heart failure patients could be influenced by heterogeneous sprouting of sympathetic nerves or alteration of $\beta$-adrenergic receptors to disrupt normal $\beta$-adrenergic receptor signalling (BARS). Because sympathetic remodelling is challenging to control experimentally, we created and validated a novel ionic model incorporating the macroscopic effects of BARS on intracellular $Ca^{2+}$ dynamics and handling for tissue-scale simulations. Using our model we created ventricular sheets with varying spatial densities and gradients to study VT behaviour, represented as a rotor. We demonstrated that our BARS model reduces action potential durations (APDs) by 16% due to drastic changes in $K^+$ and $Ca^{2+}$ regulation, consistent with experimental data from animal and human studies. We also demonstrated that the ventricular substrate with spatial BARS density $> 15\%$ for $\geq 0.1$ µM (ISO) could host faster (4.5 Hz *vs.* 3.7 Hz) and more stable (5.6 mm$^2$ *vs.* 14 mm$^2$) VTs than non-BARS substrates. We also showed that VTs hosted in substrates with spatial BARS gradients ($\geq \Delta 15\%$ could splinter into multiple slow and disorganized wavelets, indicative of fibrillation. Our model and simulation findings may help predict and enhance the effectiveness of neuromodulatory interventions for sympathetically modulated VTs.

(Received 27 May 2025; accepted after revision 13 January 2026; first published online 1 February 2026)

**Corresponding author** K. Magtibay: PhD, Université de Bordeaux, 351 cours de la Libération CS10004, Talence, 33405 Gironde, France. Email: karl.magtibay@ieee.org

## Key points

- A novel human ventricular model is used for tissue-scale simulation of sympathetic activity
- Sympathetically remodelled substrates may have fast and stable rotors
- Spatial gradients of sympathetic stimulation could break rotors into wavelets
- Simulations may help inform interventions for sympathetically modulated arrhythmias

## Introduction

The ventricles of myocardial infarction (MI) and heart failure (HF) patients are known to have heterogeneously sprouted sympathetic nerves (Clyburn et al. 2022; Franciosi et al. 2017; Herring et al. 2019) or altered sensitivities of $\beta$-adrenergic receptors (Ripplinger et al. 2016) to compensate for the loss of function, but may become arrhythmogenic. Although we understand that sympathetic hyperactivity and parasympathetic withdrawal could result in heterogeneous repolarization and conduction to support arrhythmic activity (Weperen et al. 2021), even in the absence of structural remodelling (Rosman et al. 2019), the effects of sympathetic remodelling on the spatiotemporal dynamics of ventricular tachycardia (VT) are poorly understood.

$\beta$-adrenergic stimulation (BARS) facilitates sympathetic activity in cardiomyocytes. As shown in

Fig. 1, the signalling cascade begins when catecholamines, such as epinephrine and norepinephrine, bind to $\beta$ receptors on the cell membrane and induce a conformational change in a protein. This change activates a G protein on the cytoplasmic side of the membrane, which stimulates adenylyl cyclase (AC) to convert adenosine triphosphate (ATP) into cyclic adenosine monophosphate (cAMP). Elevated cAMP levels trigger protein kinase A (PKA) to phosphorylate key intracellular proteins that regulate action potential (AP) and $Ca^{2+}$ handling, leading to increased chronotropy and inotropy (Ripplinger et al. 2016).

Sympathetic remodelling of the ventricles, exhibited by heterogeneous nerve sprouting (Zhu et al. 2022) or altered receptor spatial distribution (Beau et al. 1993; Herring et al. 2019), could create action potential duration (APD) gradients that affect VT dynamics. However controlling

for the spatial density of sympathetic elements in the ventricles may be challenging to replicate experimentally. Thus a computational model of BARS could allow us to systematically examine the role of sympathetic hyperactivity in modulating VT behaviour.

Various computational models have been developed to describe the multiple aspects of cardiac BARS (Doste & Bueno-Orovio 2021). In simpler models parameters such as maximal conductances and ion channel gating rates are adjusted to match experimental data. Although this approach is scalable to tissue- and organ-scale simulations, it captures only cellular behaviour under steady-state BARS at a single stimulation level. On the contrary complex frameworks that couple a detailed

signalling cascade model with an ionic model to capture dynamic electrophysiological changes are also available. However they are computationally expensive for simulations at scales beyond the cell.

Thus we introduce an ionic model of ventricular cardiomyocytes that incorporates intracellular dynamics driven by BARS for large-scale simulations. Using our proposed model we simulated the isolated effects of varying spatial densities of sympathetic stimulation and spatial gradients on an existing VT within non-diseased cardiac substrates with altered distributions of sympathetic elements. In the following sections we will present the derivation of our proposed model and validation procedure. Then we will outline our methods for tissue simulations,

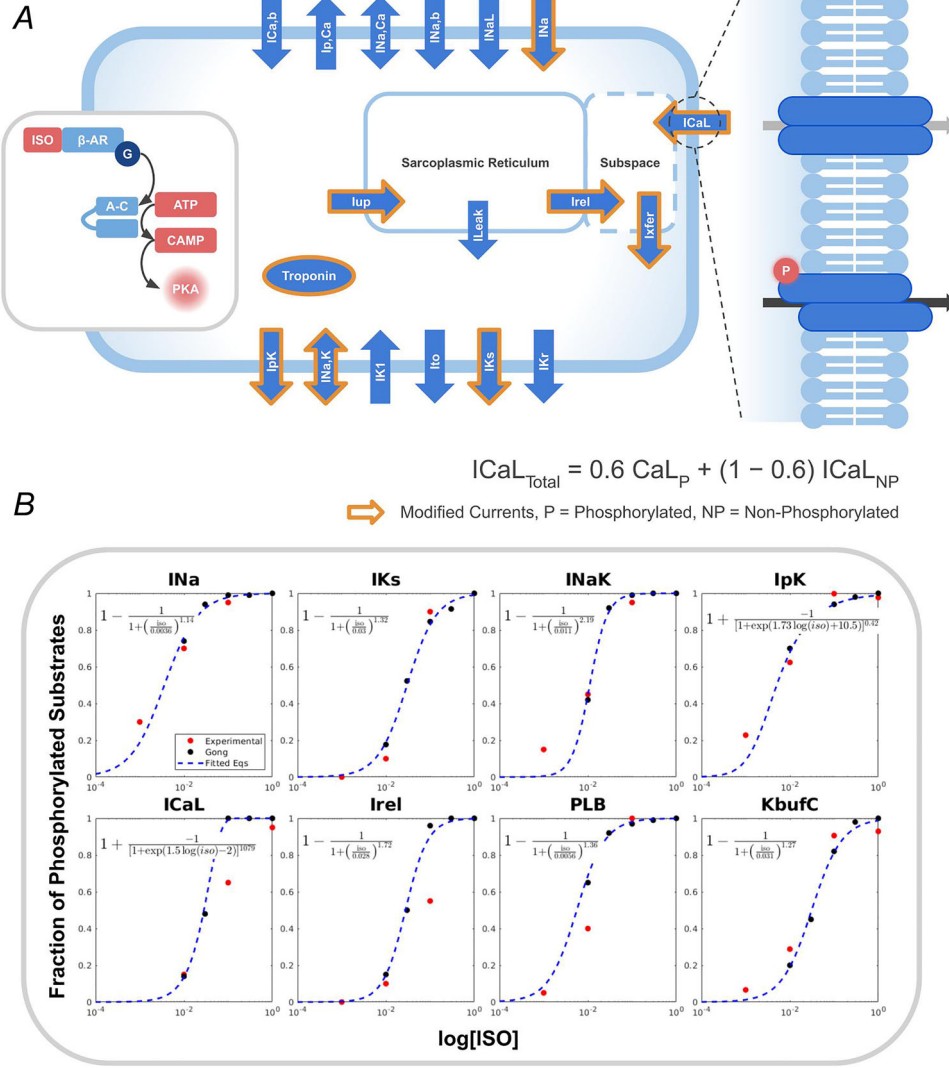

**Figure 1. Model of *β*-adrenergic receptor signalling (BARS) pathway in the cardiomyocyte**
*A*, the model captures the downstream electrophysiological effects of the BARS pathway (grey box) on the intracellular dynamics of ventricular cardiomyocytes. Modulated ionic currents and fluxes are outlined in orange. The magnified inset depicts the calculation of total currents from the fractional contribution of non-phosphorylated and phosphorylated channel populations. *B*, the BARS pathway is represented by dose–response curves relating ISO to the fractional contribution of phosphorylated channel populations.

rotor generation and VT data analysis. Subsequently we will present and discuss our results and highlight the importance of our findings for the neuromodulation of sympathetically mediated VTs.

## Methods

The following sections provide details on the creation, validation and application of our proposed ionic model. We performed all simulations described in the following sections using an open-source cardiac electrophysiology simulator (openCARP) (Plank et al. 2021; openCARP consortium et al. 2025).

### Ionic model construction

Our proposed model is based on an established ionic model of the human ventricular myocytes (TT2) (Ten Tusscher and Panfilov 2006). As shown in Table 1, we modified seven downstream current-producing channels typically phosphorylated by the BARS-cAMP-PKA cascade during sympathetic activation, along with calcium-buffering and diffusion-flux terms. Each BARS target was modelled with two sets of equations: one for its non-phosphorylated (NP) state, which is based on the original TT2 model, and another for its phosphorylated (P) state as described in the following subsections. Then, we introduced an ISO parameter to represent the concentration of a typical sympathetic agonist, iso-proterenol (ISO), used to modulate the currents and flux produced by phosphorylated intracellular channels. Logarithmic dose–response curves (Fig. 1*B*) relating ISO concentration and steady-state phosphorylation levels of target channels were fit to a combination of experimental data and data from existing BARS implementations (Gong et al. 2020).

The net current/flux of each substrate was calculated as

$$I_{net} = \alpha I_P + (1 - \alpha) I_{NP}$$

where $\alpha$ is the phosphorylation level, and $I_P$ and $I_{NP}$ are the individual currents from the phosphorylated and non-phosphorylated channels, respectively. We summarize our proposed model in Fig. 1.

### Steady-state membrane currents under *β*-adrenergic stimulation

We modified the equations of the membrane currents of phosphorylated ion channels during BARS as follows. We upregulated the maximal conductance ($G_{Na}$ by a factor of 1.7 and introduced a −5 mV potential shift to the fast inactivation gate (h) of the fast $Na^+$ channels. We also upregulated the maximal conductance ($G_{CaL}$ by a factor of 3 and shifted the voltage-dependent activation (d) and fast

(f2) and slow inactivation gates by −10 mV of the L-type $Ca^{2+}$ channels. Moreover we tuned the time constants of the d, f2 and f gates to achieve 1.25 times faster activation near threshold voltages and 1.6 times slower inactivation at highly positive potentials. We increased the maximal conductance ($G_{Ks}$ by a factor of 2.5, shifted the activation gate by −5 mV and increased the rate of activation for the slow delayed rectifier $K^+$ channels (Ks). Furthermore we scaled the intracellular $Na^+$ half-saturation constant (KmNa) of the $Na^+$-$K^+$ pump current (INaK) by a factor of 0.7 to increase $Na^+$ extrusion and prevent decreased or reversed $Na^+$-$Ca^{2+}$ exchanger function. Finally we fine-tuned the overall AP behaviour by scaling the total current of plateau $K^+$ by 2 and shifting activation by 10 mV.

### Intracellular Ca²⁺ dynamics under *β* -adrenergic stimulation

We modified the $Ca^{2+}$ handling currents by following the protocol in Gong et al. (2020). With ICaL held in its baseline state and intracellular $Ca^{2+}$ held constant, we parameterized the ryanodine receptor current, sarcoplasmic reticulum (SR) uptake current and cytoplasmic $Ca^{2+}$ buffer from their baselines due to phosphorylation. Specifically we increased the maximum conductance of the ryanodine receptor ($G_{rel}$ by a factor of 1.6, the channel's opening rate by 1.4 and closing rate by 1.8. To offset the buildup of $Ca^{2+}$ in the sarcolemmal subspace (CaSS) during an AP's plateau phase and prevent spurious CaT, the rate of $Ca^{2+}$ transfer between the subspace and bulk cytosol was also increased by a factor of 2.1, represented as the maximal conductance of $G_{xfer}$. We multiplied the half-saturation constant of the SR uptake current ($I_{up}$ by a factor of 0.85 to incorporate the prevention of phospholamban inhibition on $Ca^{2+}$ uptake, and increased the half-saturation constant for cytoplasmic $Ca^{2+}$ buffer (Kbufc) by 1.7 times to incorporate the decreased calcium affinity of Troponin I. We did not modify the maximal conductance of the $Ca^{2+}$ SR leak channel, as its phosphorylation is not directly mediated by PKA (Curran et al. 2007).

### Validation of the proposed ionic model

We performed virtual voltage-clamp experiments to validate the functionalities of our proposed ionic model. We measured peak $I_{Na}$ in 300-ms steps for potentials from −80 to 20 mV from a holding potential of 0 mV. Similarly we measured the fractions of inactivated $Na^+$ channels during the same steps for potentials from −120 to −80 mV. We measured the peak $I_{Ks}$ in 3000 ms steps for potentials between −40 and 60 mV, from a holding potential of −40 mV. We also measured the peak $I_{CaL}$ in

**Table 1. Adjusted currents and fluxes for modelling by $\beta$-adrenergic stimulation (BARS) in the cardiomyocyte**

| Parameter | Description | Change in parameters due to BARS | Reference |
|---|---|---|---|
| $I_{Na}$ | Fast Na$^+$ current | Increased channel conductance (170%) and channel inactivation shifted leftward (−5 mV) | I-V curves (canines, Baba et al. 2004); CV change (various species, Campbell et al. 2014) |
| $I_{Ks}$ | Slow delayed rectifier K$^+$ current | Increased channel conductance (250%) and channel inactivation shifted leftward (−5 mV) | APD shortening (humans, canines, Lang et al. 2015, Priori & Corr 1990) |
| $I_{pK}$ | Plateau K$^+$ current | Upregulated current (200%) and channel activation shifted rightward (+10 mV) | I-V curves (canines, Sridhar et al. 2007) |
| $I_{NaK}$ | Na$^+$-K$^+$ pump current | Decreased half-saturation constant (70%) | Na$^+$ affinity (mouse, Despa et al. 2008) |
| $I_{CaL}$ | L-type Ca$^{2+}$ current | Increased channel conductance (300%) and channel availability shifted leftward (−5 mV) | I-V curves (canines, Nagykaldi et al. 1999) |
| $I_{rel}$ | Ryanodine receptor current | Increased channel conductance (160%). Faster opening (140%) and closing (180%) rates | [Ca$^{2+}$]$_i$ and I$_{rel}$ behaviour (mice, Ginsburg et al. 2004) |
| $I_{up}$ | Sarcoplasmic reticulum uptake current | Decreased half-saturation constant (85%) | Ginsburg et al. (2004) |
| $I_{xfer}$ | Transfer from subspace to bulk cytosol | Upregulated current (210%) | Ginsburg et al. (2004) |
| $Kbufc$ | Half-saturation constant for cytoplasmic Ca$^{2+}$ buffer | Decreased affinity for intracellular Ca$^{2+}$ (170%) | Ginsburg et al. (2004) |

Currents and fluxes modulated by BARS in ventricular cardiomyocytes are modelled using data from animal and human experiments.
Abbreviations: APD, action potential durations; BARS, $\beta$-adrenergic receptor signalling; CV, conduction velocity.

500 ms steps from −40 to 60 mV, from a holding potential of −90 mV. We normalized the I-V curves obtained at ISO = 1 μM against those obtained at ISO = 0 μM. We stimulated our unit cell with a 60 μA/cm² transmembrane current at 1 Hz for 1000 beats, ensuring that the APDs and CaTs of the last 100 beats varied by less than 1%. The effects of changing ISO concentrations on APD $_{90}$ and Ca$^{2+}$ transient (CaT) amplitude were then assessed from the last beat.

Next, we assessed the rate-dependent APD $_{90}$ shortening of our model for a subset of ISO concentrations (0, 0.01, 0.1 and 1 μM). At each concentration, we ran the model for 1000 beats with pacing cycle lengths (PCLs) decreasing in 100 ms intervals from 2000 ms until loss of capture. APD $_{90}$ s were measured from the last beat of each run.

For these four ISO concentrations, we further conducted conduction velocity (CV) restitution experiments on a 1 cm cable, with 400 μm quadrilateral elements. We tuned the longitudinal ($\sigma_L$ = 0.15 S/m) and transverse ($\sigma_T$ = 0.07 S/m) conductivities of our tissue cable to achieve a CV of 60 cm/s as observed in healthy ventricular tissues, in the absence of BARS (Fu et al. 2024). We paced our cable model with 5-ms stimuli at twice the capture amplitude of our S1 stimulus at 1 Hz for 20 beats to propagate a wave from left to right. We delivered an S2 stimulus with incremental CLs from 250 to 1000 ms. For each CL increment, we calculated CV as the difference in activation times along the 0.25 cm and 0.75 cm portions of our cable.

### Applications to tissue-level simulations

We applied our proposed ventricular ionic model to simulate the effects of BARS spatial density on a VT in two-dimensional substrates.

**Two-dimensional substrates, BARS spatial density and rotor generation.** We paced a cell using a 60 μA/cm², 0.5 ms transmembrane current, with a 600 ms basic CL over 60 s to reach a steady state. Then, we used the cell's state variable values at rest as initial conditions to adjust the longitudinal ($\sigma_L$ = 0.1608 S/m) and transverse ($\sigma_T$ = 0.0448 S/m) conductivities of a cable to match a physiological ventricular tissue with approximately 60 cm/s CV (Fu et al. 2024). We created a 36 cm² (6 cm × 6 cm) square and a 54 cm² (9 cm × 6 cm) rectangle sheet with a spatial resolution of 400 μm quadrilateral elements. Our substrates have unidirectional fibre directions to minimize the effect of their orientation on a rotor's trajectory.

We designated random elements of the square sheet to be adrenergically stimulated, simulating sympathetic remodelling of non-diseased, two-dimensional (2D) cardiac substrates due to heterogeneously sprouted nerves. We gradually increased the spatial density of BARS from 5% (2 cm²) to 50% (18 cm²) in increments of 5%, as shown in Fig. 2*A* and as in the distribution of sympathetic nerves in the human ventricles (Kawano et al. 2003).

We divided our rectangular sheet into three even regions. We started the spatial density of BARS at 20% (3.75 cm²) on the entire sheet, then gradually decreased and increased their BARS density by 5% (0.75 cm²) on left and right regions (i.e. from Δ 5% to Δ 20%), respectively, as shown in Fig. 2*B*, as in the redistribution of sympathetic elements due to HF (Beau et al. 1993). We repeated the random distribution of BARS elements 10 times in a square (*n* = 10) and a rectangular (*n* = 10) sheet, each iteration with a different ISO concentration (0.01, 0.1 and 1 μM), to simulate the degrees of sympathetic activation, and a different random number generator seed to represent a unique substrate.

We used a monodomain framework to simulate wave propagation in ventricular tissues. Transmembrane currents in a monodomain model (Clayton et al. 2011; E. J. Vigmond et al. 2002) are expressed as functions of transmembrane voltage ($\nabla\mathbf{V}_m$, ionic ($\mathbf{I}_{ion}$ and capacitive currents ($\mathbf{C}_m\frac{\partial\mathbf{V}}{\partial\mathbf{t}}$ over a cell membrane, and the external current stimulus ($I_{stim}$ applied to a cell with a surface-to-volume ratio ($A_m$. Moreover the extracellular conductivity tensor ($\sigma_e$ could be expressed as a scalar proportion of the intracellular conductivity tensor ($\sigma_i$ such that $\sigma_e = \lambda\sigma_i$. Thus, a monodomain equation could be written as

$$\frac{\lambda}{\lambda + 1}\nabla \cdot (\sigma_i\nabla V_m) = A_m\left(C_m\frac{dV_m}{dt} + \Sigma I_{ion} - I_{stim}\right) \quad (1)$$

We created rotors using the S1–S2 method (Feola et al. 2017) to simulate a VT. Using the steady-state intracellular parameters and tuned conductivities of our ionic model and meshes, we paced our substrates at their leftmost edge using a 2 ms, 250 μA/cm³, transmembrane current stimulus with 600 ms BCL to create a uniform wavefront. Then using the same transmembrane current activation, we stimulated the lower half of the square substrate perpendicular to the travelling wave during the repolarization of its right half. Figure 2*C–F* summarizes our rotor generation process. We generated rotors across all random seeds, BARS density and gradients. We simulated rotors for 5 s.

**Rotor analysis.** We transformed our AP signals to instantaneous phase signals using the Hilbert transform (Kuklik et al. 2014) to identify and study the changes in the dynamics of rotor phase singularities due to BARS density. We band-pass filtered (0.1–15 Hz) our AP signals to

extract sinusoid-like traces ($\mathbf{x}(t)$ and obtain their Hilbert transform ($\mathbf{H}[f]$, such that

$$H\left[f\right] = H\left[x\left(t\right)\right] = \frac{1}{\pi}\, p.v. \int_{-\infty}^{+\infty} \frac{x\left(t\right)}{t-\tau}d\tau \qquad (2)$$

where *p.v.* is the Cauchy principal value that is used to correct the values of $\mathbf{H}\{\mathbf{x}(t)\}$ when $t = 0$ (Kuklik et al. 2014). Then $\mathbf{H}\{\mathbf{x}(t)\}$ could be written as a complex expression, $\mathbf{h}(t) = \mathbf{h}(t)_r + j\mathbf{h}(t)_i$ from which an instantaneous phase ($\theta(t)$ could be calculated with $\theta(t) = \tan^{-1}[\mathbf{h}(t)_i / \mathbf{h}(t)_r]$.

We designated $-\pi/2$ the wavefront of a rotor since its natural incorporation when solving for the analytic signal $\mathbf{h}(t)$ via Hilbert transform (King et al. 2017). We identified wavefront phase singularities using a local linear regression method on a $10 \times 10$ grid to identify all phase values (Grey et al. 1998).

We analysed the changes in a rotor's angular speed (Hz) and localization area (mm$^2$) as indicators of spatiotemporal behaviour due to increasing BARS density.

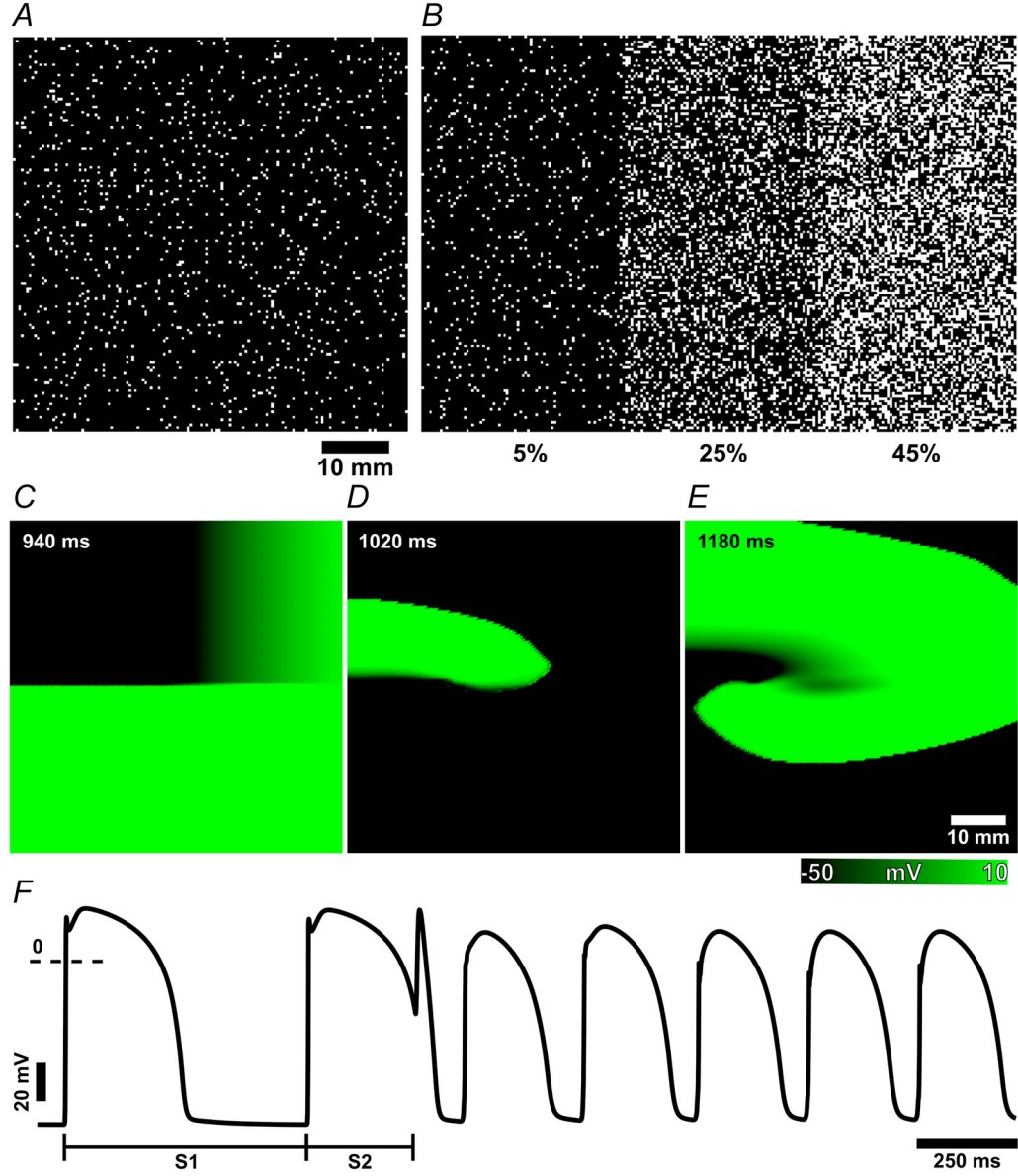

**Figure 2. Square and rectangular ventricular sheets with β-adrenergic site distribution and rotor generation using the S1–S2 protocol**
*A*, an example of a square sheet with 5% β-adrenergic stimulation (BARS) density indicated as white dots. *B*, an example of a rectangular sheet with different BARS density. *C–E*, an area perpendicular to the travelling wave could be stimulated during repolarization to initiate a rotor. *F*, an example of action potential trace showing the S1–S2 stimuli followed by a signal pattern indicating the presence of a rotor.

We calculated a rotor's angular speed from the number of revolutions completed of a fixed point 4 mm away (typical interelectrode distance of a grid mapping catheter (Deno et al. 2020)) from the location of a verified phase singularity over at least 4 s. Moreover we measured a rotor's spatiotemporal organization by calculating the total substrate area over which a phase singularity was present for 4 s. A meandering phase singularity may cover a large substrate area, whereas a stable phase singularity would cover only a small area.

**Statistical Analysis.** Due to the hierarchical and repeated nature of our data (i.e. each substrate has 3 ISO concentrations and each ISO concentration has 10 spatial BARS densities or 4 spatial gradients), we performed a generalized linear mixed effects model (GLMM) with a linear identity link to analyse the effect of BARS spatial density on a rotor's angular speed and localization. Post hoc $F$ tests were performed to assess the overall impact of increasing BARS density on each rotor characteristic with a significance level of 5% ($\alpha = 0.05$. All statistical analyses were performed using SPSS, a proprietary statistical package from IBM (George & Mallery 2019).

## Results

### Validation of outputs of the new ionic model

In Fig. 3 we present steady-state traces for the eight BARS substrates generated by the model under 1 Hz pacing for

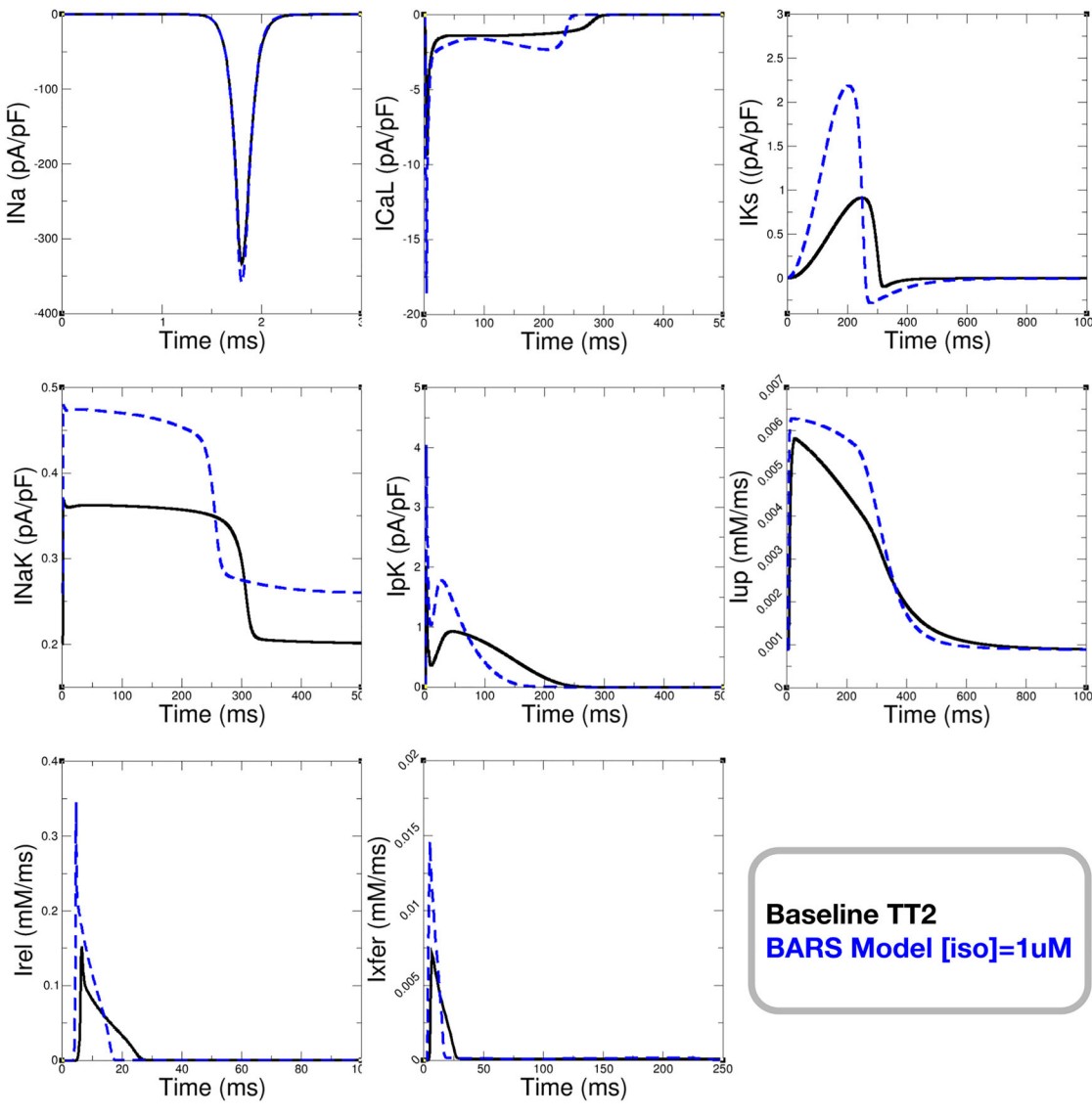

**Figure 3. Ionic currents and fluxes under BARS**
*A*, fast Na$^+$; *B*, L-type Ca$^{2+}$; *C*, slow delayed rectifier; *D*, Na$^+$-K$^+$ exchanger; *E*, plateau K$^+$; *F*, SERCA uptake; *G*, Ca$^{2+}$-induced Ca$^{2+}$ release and *H*, diffusive flux of Ca$^{2+}$ from subspace to bulk cytoplasm at steady state

ISO = 0 μM and ISO = 1 μM. The channel dynamics for select ion channels, changes in overall $Ca^{2+}$ handling and APs are subsequently included in Fig. 4.

We show in Fig. 3*A* that increasing the maximal conductance of phosphorylated $I_{Na}$ (G $_{NaP}$ results in a larger current amplitude (−358.2 *vs.* −332.9 pA/pF) and a 1.5-fold increase in peak current density of the I-V curve, above the range reported in Baba et al. (2004) (1.32 ± 0.052-fold). Additionally although inactivation gating is accordingly shifted in our model, no modifications are included for the activation gate as we found that incremental negative shifts in the half-activation value produced exaggerated CV increases. The discrepancies between the model modifications and experimental data were adopted so that the resulting CV increase remains in line with the estimated impact from $I_{Na}$ alone (Campbell et al. 2014).

We illustrate in Figs. 3*B* and 4*B* the effects of $I_{CaL}$ modifications due to BARS. We show that the peak current density of the I-V curve for phosphorylated $I_{CaL}$ is increased 3.6-fold from its baseline at −10 mV, consistent with experimental findings by Nagykaldi et al. (1999), Ganesan et al. (2006) and Antoons et al. (2007). The peak current amplitude increases (−18.6 *vs.* −9.5 pA/pF) and the AP plateau phase is shortened. We also observed the appearance of a slight crest during the AP's plateau phase.

We present in Fig. 3*C* a significant increase in $I_{Ks}$ amplitude (2.2 *vs.* 0.93 pA/pF). In Fig. 4*C* we present a 2.3 times increase in $I_{Ks}$ I-V peak current density for a 3000 ms duration clamp pulse, within the range of experimental data (Volders et al. 2003).

We depict in Fig. 3*E* the upregulation of $I_{NaK}$ by a factor of 1.3 as a result of BARS, concordant with the percentage changes experimentally measured in previous studies (Despa et al. 2008; Gao et al. 1992; Zhang et al. 2006). We also depict in Figs 3*G–I* a 2.3 times increase in $I_{rel}$ amplitude, 1.9 times increase in $I_{xfer}$ and 1.1 times increase in $I_{up}$ respectively. The resulting intracellular CaT peak (Fig. 4*D*) increased by a factor of 2.25 and duration decreased by 0.21, which are comparable to experimental data presented by (Yamada & Corr 1992). Additionally we find that zeroing $I_{NaK}$ resulted in a 1.5 times increase in CaT. Despite differences in $Ca^{2+}$ handling mechanisms, these trends are consistent with observations in murine cardiomyocyte models (Despa et al. 2008; Kuzumoto et al. 2008).

Overall we present in Fig. 4*E* the profile of a steady-state AP from our proposed model under BARS compared to its baseline. The macroscopic effects of the current modifications include a faster upstroke velocity (129.2 mV/ms *vs.* 126 mV/ms), larger peak (43.6 mV *vs.* 40.6 mV) and phase 2 (34.9 mV *vs.* 26.5 mV) potential, and shorter APD (260 ms *vs.* 310 ms).

## Restitution results

Our APD restitution shows that as ISO concentration increases, CaT amplitude and AP shortening increase non-linearly, as seen in Fig. 5*A* and *B*. Between the baseline and 0.01 μM, the CaT amplitude increases by a factor of 1.25. However the combined effect of a shortened CaT duration and an increase in repolarizing IKs resulted in a decrease in APD. Between 0.01 μM and 0.1 μM ISO, the increased CaT growth is further countered by the increased phosphorylated channels of repolarizing currents, resulting in significant APD shortening. At 1 μM ISO the fraction of phosphorylated substrates is maximized, resulting in plateaued CaT and APD at

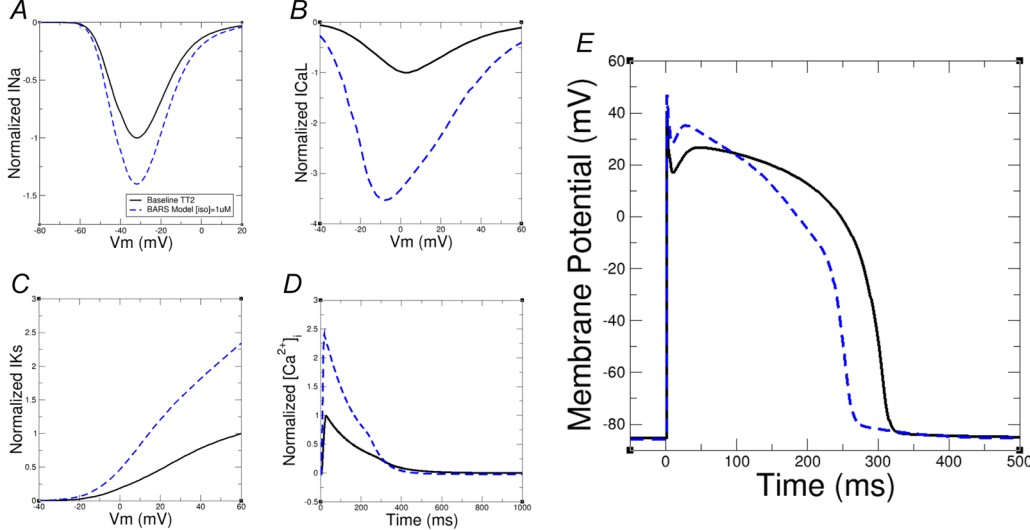

**Figure 4. Channel kinetics and $Ca^{2+}$ handling under BARS**
*A–C*, simulated I-V curves; *D*, steady-state $Ca^{2+}$ transients; *E*, steady-state Aps

2.25 and 0.83 times the baseline values, respectively. The fold change in CaT and APD produced by our model is within ranges observed from other BARS implementations (Gong et al. 2020; Heijman et al. 2011) and experimental data (Lang et al. 2015; Priori & Corr 1990).

In Fig. 5*C*, we further demonstrate the model's rate-dependent APD shortening. ISO led to APD shortening across all PCLs, with larger relative effects at longer PCLs. As ISO increases, the model successfully produces APs at shorter PCL lengths, resulting in a greater difference between the shortest and longest APDs.

Finally we show the results of our CV restitution study in Fig. 5*D*. We report an increase in the maximum CV from 60 to 61 cm/s due to 0.01 µM ISO, and a decrease in the minimum CV from 47 to 41 cm/s, at DIs of 375 and 325, respectively. Increasing the ISO concentration to 0.1 µM produces a steady-state CV of 62 cm/s and decreases the minimum CV to 35 cm/s at a DI of 275 ms. Although we did not observe significant changes to steady-state CV between 0.1 and 1 µM ISO, the minimum CV decreases further to 30 cm/s at a DI of 250 ms.

## Tissue simulations

The following sections present our measurements and analysis of rotors hosted in sympathetically remodelled substrates.

**Rotors in ventricular substrates with BARS.** Rotors hosted on ventricular substrates with BARS have faster angular speeds and are more spatiotemporally organized than those hosted on plain substrates. We show in Fig. 6 isometric views of the trajectory of phase singularities over time for three BARS density levels (i.e. 5%, 10% and 15%).

We show in Fig. 7*A* and Supplemental Video S1 that rotors hosted in uniformly BAR-stimulated substrates (Full) have faster angular speeds at higher ISO concentrations than plain substrates (i.e. $3.7 \pm 0.0..$ Hz [0.0 µM] $< 4.2 \pm 0.0..$ Hz [0.01 µM] $< 4.5 \pm 0.0..$ Hz [0.1 µM] $= 4.5 \pm 0.0..$ Hz [1.0µM], $P < 0.001$). Although slower than those hosted on homogeneous substrates, we also show that the angular speed of hosted rotors increases with higher ISO concentrations, regardless of the degree (Any % BARS) of sympathetic remodelling

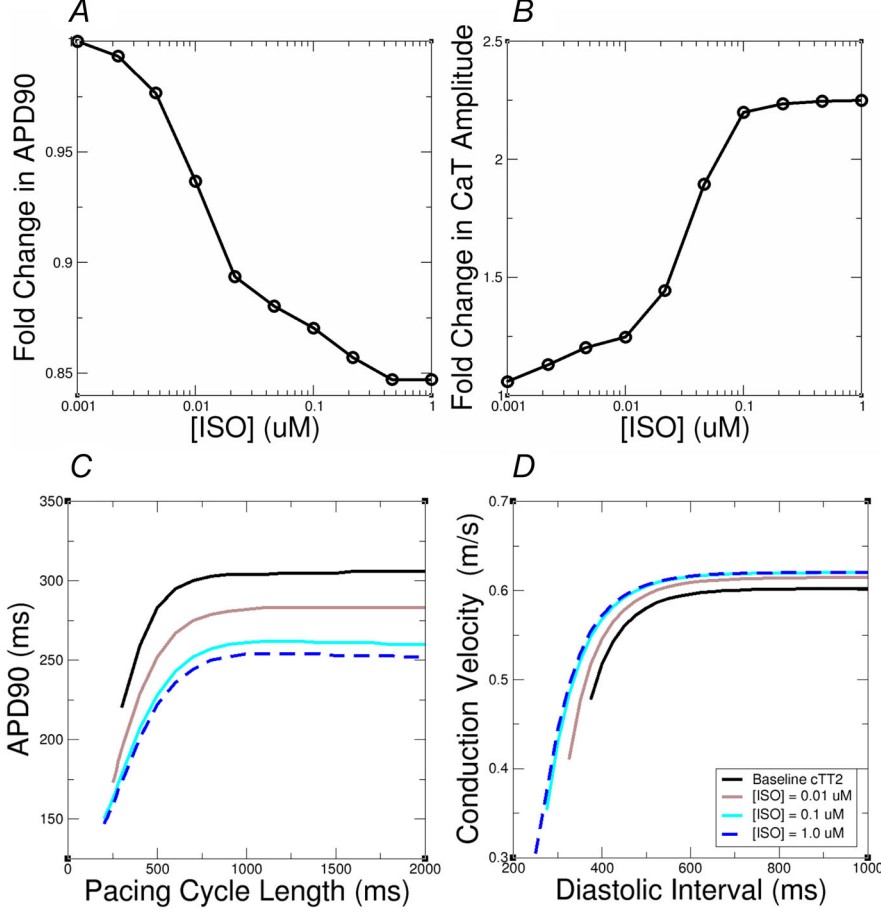

**Figure 5. BARS dependence of APD, CaT and CV**
*A*, APD at 90% repolarization (APD $_{90}$; *B*, intracellular Ca$^{2+}$ transient amplitude; *C*, APD $_{90}$ rate dependence; *D*, CV restitution

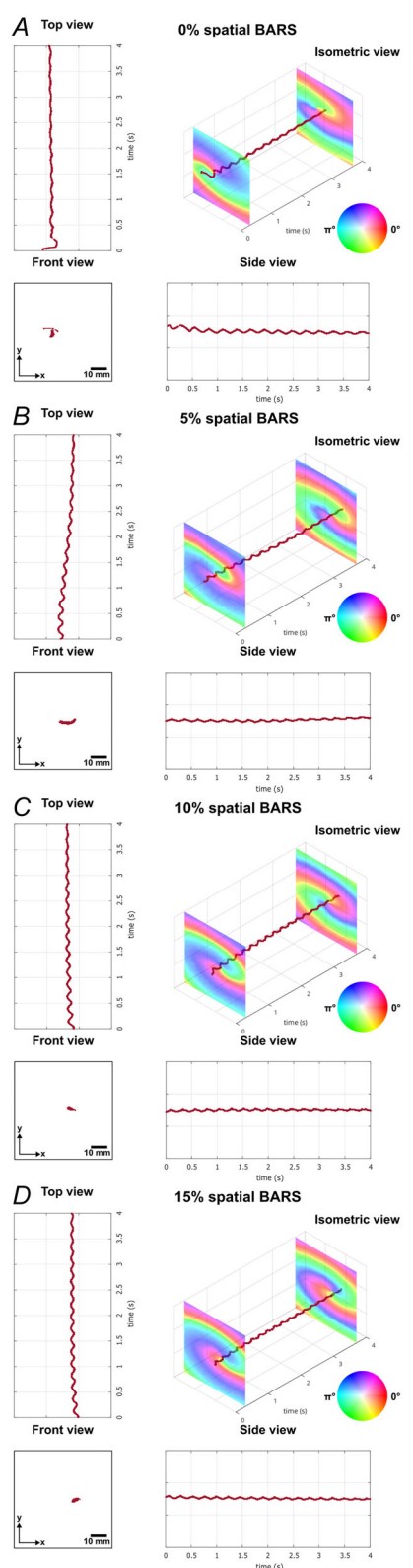

with a slowly rotating rotor. Increasing spatial BARS density (*B* and *C*) could stabilize the behaviour of a phase singularity and create faster rotating rotors.

---

(i.e. 4.0 ± 0.03 Hz [0.01 μM] < 4.2 ± 0.03 Hz [0.1 μM] < 4.4 ± 0.03 Hz [1.0μM], *P* < 0.001). A more detailed examination reveals that a rotor's angular speed gradually increases for ventricular substrates with spatial BARS densities ≤ 15% (from 3.9 ± 0.04 Hz to 4.2 ± 0.04 Hz). Rotor angular speeds minimally change in substrates with spatial BARS densities >15% regardless of the ISO concentrations administered (Fig. 7*B*).

Similarly rotors become spatiotemporally organized in uniformly BAR-stimulated substrates, as indicated by the significant narrowing of their localization areas compared to plain substrates (14.1 ± 0.0.. mm² [0.0 μM] >> 4.5 ± 0.0.. mm² [0.01 μM] < 5.6 ± 0.0.. mm² [0.1 μM] = 5.6 ± 0.0.. mm² [1.0 μM], *P* < 0.001), as we illustrated in Fig. 7*C* and Supplemental Video S1. In contrast we show that sympathetic remodelling and increased ISO concentrations slightly disrupt a rotor's spatiotemporal organization (i.e. 7.7 ± 2.1 mm² [0.01 μM] < 9.4 ± 2.1 mm² [0.1 μM] < 11.0 ± 2.1 mm² [1.0 μM], *P* < 0.001). Nonetheless sympathetically remodelled ventricular substrates with spatial BARS densities greater than 15% could support spatiotemporally organized rotors (> 10.5 ± 3.8 mm², *P* < 0.001) regardless of ISO concentration (Fig. 7*D*).

**Effect of sympathetic spatial gradients on rotors.** Ventricular substrates with sharp spatial gradients of sympathetic remodelling could help create slower and spatiotemporally disorganized wavelets from a primary rotor, across all ISO concentrations, as illustrated in Fig. 8 and Supplemental Video S2. Specifically wavelets could be splintered from the primary rotor at the boundary where the APD and CV gradients are the greatest. For BAR-stimulated substrates with 0.01 μM ISO, wavelets began to form only after a rotor had been sustained (Fig. 8*A*). In contrast primary rotors are immediately splintered to wavelets in BAR-stimulated substrates with ISO concentrations ≥ 0.1 $\mu$ M (Fig. 8*B* and *C*).

We show in Fig. 7*E* that a rotor's tail could break into several wavelets in substrates with ≥ Δ15% spatial BARS gradients as indicated by the average number of detected phase singularities across all ISO concentrations (2.0 ± 0.4 for Δ15% and 4.1 ± 0.4 for Δ20% *P* < 0.001). Moreover the angular speed of wavelets formed in sheets with Δ20% spatial BARS gradients is faster (0.8 ± 0.7 Hz *vs.* 1.8 ± 1.0 Hz, *P* < 0.001) and spatiotemporally organized (100.7 ± 97.0 mm² *vs.* 413.4 ± 60.8 mm², *P* < 0.001) than those formed in sheets with Δ15% as shown in Fig. 7*F* and *G*, respectively.

**Figure 6. Trajectory and organization of phase singularities could be influenced by the spatial density of $\beta$-adrenergic stimulation (BARS)**
Phase singularities hosted in slightly sympathetically dense (*A*) substrates may migrate from their starting point and be associated

Similarly among the three ISO concentrations examined, wavelets are slower (1.1 ± 0.7 Hz < 4.2 ± 0.2 Hz [0.01 μM], 2.1 ± 1.2 Hz < 4.4 ± 0.4 Hz [0.1 μM] and 2.1 ± 1.1 Hz < 4.8 ± 0.6 Hz [1.0 μM]) and more spatiotemporally disorganized (28.2 ± 12.0 mm² > 6.0 ± 3.4 mm² [0.01 μM], 643.4 ± 588.9 mm² > 21.1 ± 8.9 mm² [0.1 μM] and 555.7 ± 396.1 mm² > 12.2 ± 4.1 mm² [1.0 μM], $P < 0.001$) than their primary rotors for all spatial BARS gradients as shown in Fig. 7*H* and *I*, respectively.

## Discussions

We introduced a novel ionic model of the human ventricular cells, incorporating intracellular $Ca^{2+}$ dynamics due to BARS for tissue-level simulations. Various computational models have been developed to investigate the functional effects of BARS on cardiac electrophysiology. However early studies (Kuzumoto et al. 2008; Zeng & Rudy 1994) mainly involve scaling the magnitudes or shifting the activation/inactivation curves of the ionic currents targeted by the adrenergic response. Although computationally efficient, these models only capture ADP shortening and $Ca^{2+}$ handling upregulation at steady-state under maximal BARS. They are thus limited for studying how different levels of stimulation can affect EP properties. Models developed by Greenstein et al. (2004) and Terrenoire

et al. (2005) incorporate variable BARS by summing the fractional current contributions from phosphorylated and non-phosphorylated substrates, a method similar to ours. Yet only the L-type $Ca^{2+}$ channels and the slow rectifier $K^{+}$ channels in these two models, respectively, are formulated with dual populations; other BARS substrates are still scaled. Substantially more complex frameworks (Gong et al. 2020; Heijman et al. 2011; Saucerman et al. 2003) couple a full signalling cascade with an EP model, thereby capturing the detailed temporal responses of multiple interacting BARS pathways. Given the high computational cost associated with solving the full set of equations, these frameworks are typically not applied directly in large-scale simulations. Rather 3D geometric models can be initialized using the state variables of the coupled cell model after an appropriate number of beats at the BARS level of interest. However implementing this approach remains computationally burdensome in practice, particularly for simulations with innervation gradients or heterogeneity, where each different level of adrenergic stimulation would require separate cell-level runs and initialization steps.

Building upon these existing studies we capture the non-linear effects of BARS signalling on the ionic model as a set of logarithmic dose–response functions, providing a continuous mapping between ISO concentration and the phosphorylation level of a target channel. This

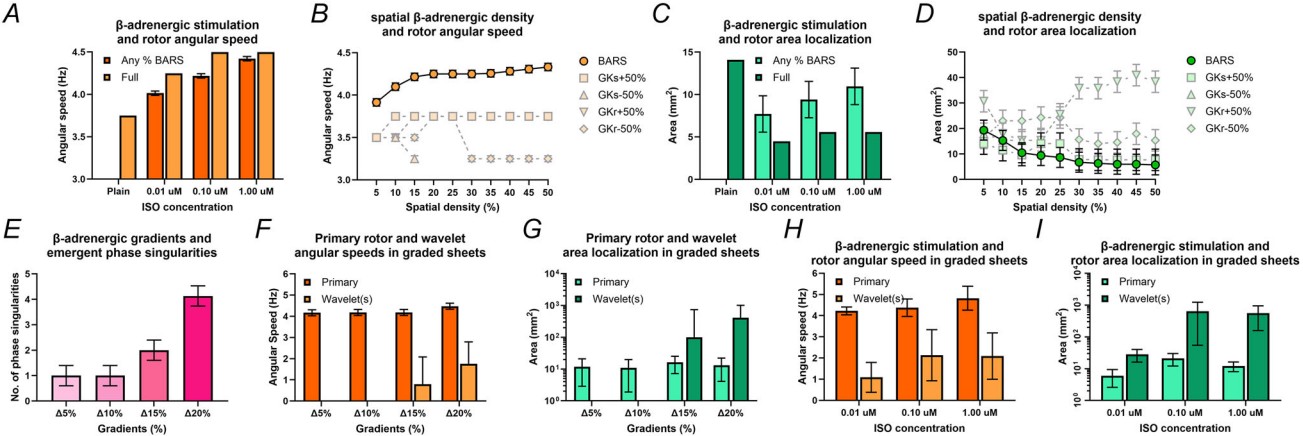

**Figure 7. Changes to rotor angular speed and localization due to sympathetic remodelling in ventricular substrates**

*A*, changes to a rotor's angular speed due to different degrees (spatial BARS % of substrate and ISO concentrations) of sympathetic stimulation of ventricular substrates. Any % BARS indicates the average measured value from 5% to 50% spatial BARS, whereas Full indicates 100% spatial BARS or a homogeneous BARS substrate. *B*, increasing spatial BARS densities also increases a rotor's angular speed, which is maintained at higher degrees (> 15%. Compared to other APD modifiers, rotors hosted in BARS substrates have faster angular speeds. *C*, broad areas of rotor localization in plain substrates indicate meandering phase singularity, hence great spatiotemporal disorganization. However, BARS substrates could host spatiotemporally organized rotors. *D*, substrates with low spatial BARS densities are more likely to host meandering rotors, while substrates with spatial BARS densities > 15% could host stable rotors. *E*, introducing spatial BARS gradients ≥ 15% could splinter a rotor into multiple slower wavelets, which are slower *F*, and more spatiotemporally disorganized *G*, than their primary rotor across all ISO concentrations. Finally regardless of ISO concentrations, wavelets maintain their angular speed *H*, and spatiotemporal *I*, characteristics relative to their primary rotors for all spatial BARS gradients.

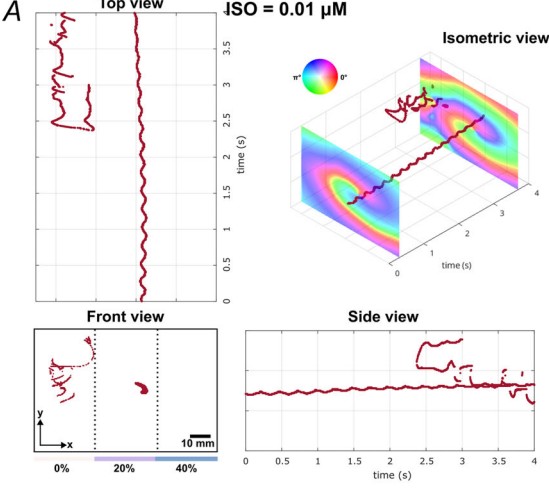

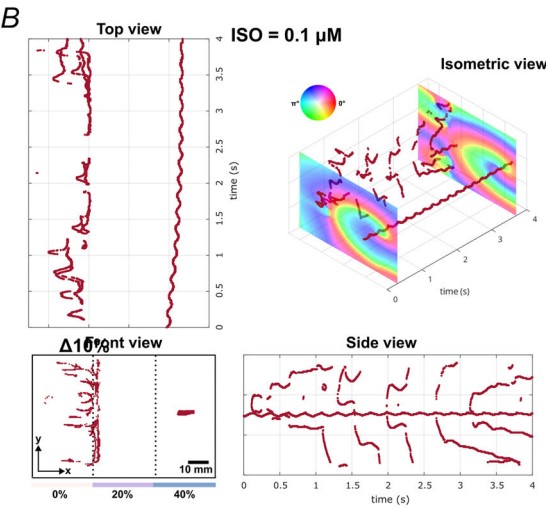

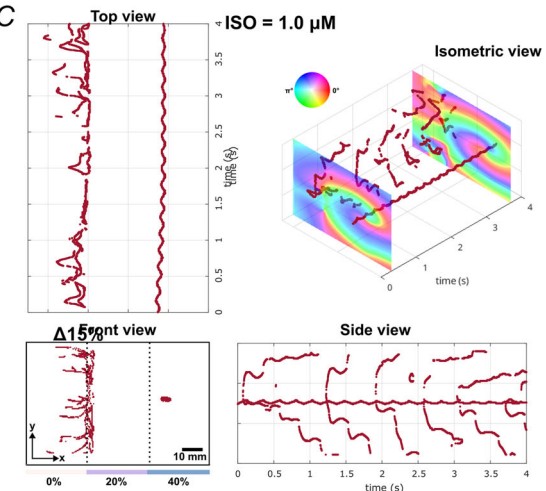

**Figure 8. Trajectory and organization of phase singularities of primary rotors and their wavelets are influenced by spatial *β*-adrenergic stimulation (BARS) gradients and ISO concentrations**

*A*, 0.01 µM; *B*, 0.1 µM; and *C*, 1.0 µM.

eliminates the need to rerun a stand-alone cell model each time phosphorylation information is required for a different stage of the BARS cascade. Our framework provides practical advantages for dissecting how spatially heterogeneous BARS responses can contribute to arrhythmogenic propensity at the tissue and organ level, a relationship which may be challenging to investigate experimentally.

We selected the TT2 model as our baseline due to its computational efficiency and extensive history of integration into multiscale arrhythmia simulations (Prakosa et al. 2018; Shade et al. 2021; Zhang et al. 2023). Although more physiologically realistic representations of human ventricular EP, such as the ToR-ORd model (Tomek et al. 2019), have since been developed, a recent benchmarking study (Dong et al. 2025) underscores several key strengths of TT2. In isotropic 2D slabs TT2 demonstrated stable reentrant behaviour across a wider range of time steps, while requiring substantially shorter simulation times than the ToR-ORd model. The average mesh resolution for these studies was relatively fine (averaging 116–231 µm), suggesting that for more mechanistically detailed cell models, the choice of element size must also be carefully tuned to balance solution stability against computational cost. Similarly 3D simulations using TT2 did not produce any non-clinical reentries associated with numerical instability, whereas ToR-ORd occasionally generated 'pseudo-reentries' with the largest time step. Taken together these considerations make a TT2-based BARS model well suited for investigating how heterogeneous BARS stimulation affects arrhythmia dynamics. The model offers a robust and efficient foundation that is less sensitive to time step and mesh resolution, thereby ensuring that observed wavefront propagation patterns more reliably reflect physiological mechanisms rather than numerical artefacts.

Our findings from tissue-level simulations illustrate the potential utility of our novel ionic model, especially for sympathetically remodelled substrates and sympathetically mediated VTs. We demonstrated that substrates with increased ISO concentrations could support rotors with faster angular speeds and greater spatiotemporal organization than those hosted in plain substrates. This finding is consistent with our restitution curves, which indicate the shortening of APD $_{90}$ (Fig. 5*C*) and increased CV (Fig. 5*D*), indicating that substrates with overall shorter repolarization periods can anchor rotors with faster angular speeds.

Uniformly BAR-stimulated substrates could support rotors with faster angular speeds and greater spatiotemporal organization than those hosted on sparsely BAR-stimulated substrates. Nonetheless a substrate's spatial BARS density could modulate a rotor's angular speed and spatiotemporal organization. Specifically

rotors could become fast and anchored in ventricular substrates with >15% sympathetically stimulated elements due to their shortened ERPs, creating highly organized rotational activity. We compared our findings with other known APD modifiers to evaluate whether they are uniquely attributable to BARS' multi-target activation or could be replicated with a single target, such as varying the conductances of slowly ($G_{Ks}$ and rapidly ($G_{Kr}$ activating delayed rectifier potassium currents. We increased and decreased the values of $G_{Ks}$ and $G_{Kr}$ by 50%, respectively, and gradually increased their spatial densities, similar to what we did with adrenergically stimulated substrates. We showed in Fig. 7*B* that rotors hosted in sympathetic-remodelled are faster compared to those hosted in substrates with only $G_{Ks}$ or $G_{Kr}$ modified. Although rotors in substrates with increasing spatial densities of $G_{Ks}$ +50% elements have similar spatiotemporal organization as those hosted in adrenergically stimulated substrates, rotor trajectories are disrupted in substrates with other APD modifiers (i.e. $G_{Ks}$ +50%, $G_{Kr}$± 50%) as we showed in Fig. 7*D*. Interestingly we observed that rotors self-terminate only in substrates with > 15% spatial densities of elements with $G_{Ks}$ −50%. Our observations suggest that rotor dynamics in BAR-stimulated substrates may be better linked to the multi-target modulation by ISO than to simple APD modification and heterogeneity.

Sustained and organized rotors from our simulations are consistent with rotors observed in animal and *in silico* models of sympathetically stimulated ventricles (Kato et al. 2012). More importantly we showed that a 15% spatial BARS density may be a critical threshold for ventricular substrates to host rapid, anchored VTs that may require cardioversion to terminate. Our findings may explain previous clinical reports of VT patients with internal cardioverter defibrillators (ICD) who developed fast, durable VTs during sympathetic overdrive (Lampert et al. 2000).

Our simulations indicate that when substrates have sharp gradients of spatial BARS densities, a condition that may be observed after the sympathetic remodelling of the non-diseased ventricular substrate due to scarring from a distant region, the wavetail of a rotor could break into several slower wavelets at the border of areas with the greatest APD and CV difference, consistent with previous experimental and computational studies (Kato et al. 2012; E. Vigmond et al. 2002). We showed that such a gradient could be achieved by either a spatial BARS gradient or by the degree of sympathetic activity, simulated via different ISO concentrations. More importantly our tissue simulations suggest critical thresholds for sympathetic spatial gradient, $\geq \Delta 15\%$ and ISO concentration, $\geq 0.1\ \mu$M, may produce several slowly meandering wavelets and could be stabilized at the border of an area with the greatest APD and CV difference. At the same time their

primary rotor's initial characteristics are maintained. Our findings from the tissue simulations are consistent with those from the cell-level simulations; as we showed in Fig. 5*C* and *D*, there is little difference between the restitution curves of the cells stimulated with 0.1 and $1.0\mu$ M ISO and as the overall APD and CV between consecutive spatial BARS densities > 15% are similar to each other, which may explain why we did not observe splintering of rotor wavetails at the other gradient border. Our simulations suggest that a VT may eventually degenerate into fibrillation when hosted in a substrate with a sympathetic spatial gradient within a critical range, consistent with the mother rotor hypothesis of fibrillation (Jalife et al. 2002; Xu et al. 2023).

## Limitations

Computational modelling of BARS presents significant challenges due to the system's multilayered and compartmentalized regulation of cardiac EP. Our approach necessarily simplifies several aspects, most notably in the representation of the BARS pathway as a set of steady-state dose–response curves, which are unable to capture the transient effects of ISO. Adrenergic surges, which have been shown to promote arrhythmogenesis by disrupting the balance of inward and outward currents (Dajani et al. 2023; Xie et al. 2014), cannot be directly studied by the current model. Future extensions of the framework could include the time course of phosphorylation for the various target channels. We also do not incorporate adrenergic receptor isoforms (i.e. $\beta_1$ $\beta_2$ and $\alpha_1$, their localization in functional intracellular domains (e.g. subspace, caveolae and cytosol) of the myocyte or the impact of changes in downstream signalling proteins. We omit other domain-specific signalling cascades that target the same PKA substrates, such as the ryanodine receptor, which are regulated by calmodulin kinase. Consequently the model does not characterize the synergistic effects of these two pathways on the CaT. Additionally the model does not account for feedback mechanisms that lead to BARS desensitization. Therefore, studies using prolonged model stimulation should be interpreted with caution.

We recognize that the spatial resolution of our sheets is coarse, which limits their ability to represent the adrenergic stimulation of a cardiomyocyte accurately. Tissue- and organ-level simulations typically treat cardiac substrates as a syncytium; thus each mesh element in a model represents a collection of cardiomyocytes (Clayton et al. 2011; Plonsey 1988). Hence we modelled the distribution of adrenergically stimulated cardiomyocytes as area proportions to be consistent with histological evidence (Kawano et al. 2003) and to adequately represent their macroscopic effects on existing VTs.

We further acknowledge that we simulated rotors only in sheets, which do not account for the complex trajectory and spatiotemporal evolution of meandering rotors in three-dimensional ventricular substrates (Nair et al. 2011). Moreover rotors could spawn multiple phase singularities, each with its unique angular speed and organization, in the presence of structural heterogeneity (Jalife et al. 2002; Xu et al. 2023). Despite the absence of complex three-dimensional structures, we demonstrated the expected rotor behaviour in functionally remodelled ventricular substrates.

Finally we tracked the number of completed rotor rotations from a fixed point, 4.0 mm from the location of an identified phase singularity, to calculate its angular speed. Although 4.0 mm is the average interelectrode distance of a grid mapping catheter widely used in catheterization clinics today, other grid mapping catheters with different interelectrode distances are available. In the context of our present study the measured angular speeds would vary depending on the choice of reference point: closer locations would yield faster angular speed measurements, and farther locations would yield slower measurements. Thus, we must consider our relative observation points when studying rotor dynamics.

## Conclusions

We developed a novel ionic model of the ventricular myocyte to capture the macroscopic effects of the complex intracellular dynamics associated with sympathetic remodelling in MI and HF patients. Our proposed model allows for a varied range of ISO concentrations to replicate the electrophysiological response of ventricular myocytes to BARS. Furthermore we used our model to build ventricular sheets with varied spatial BARS densities to study the effect of sympathetic remodelling on VT dynamics. Ventricular substrates with a spatial BARS density greater than 15% could help anchor fast, spatiotemporally stable rotors. Moreover spatial BARS gradients $\geq \Delta 15\%$ in sympathetically remodelled substrates could break rotors into wavelets indicative of VT degradation to ventricular fibrillation. Our novel model could help inform cardioneuroablation strategies for targeting sympathetically modulated VTs. Our computational framework could be extended to enhance neuromodulatory intervention planning by predicting how vagal stimulation, ganglia ablation or pharmacological modulation impacts VT dynamics.

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

## Additional information

### Data availability statement

Raw data are available from the corresponding author upon reasonable request.

### Competing interests

The authors declare no conflicts of interest.

### Author contributions

**K.Z**. and **K.M**.: conceptualization, data curation, formal analysis, investigation, methodology, validation, visualization, writing – original draft, writing – review and editing. **N.T**. and **E.V**.: funding acquisition, project administration, resources, software, supervision, validation, writing – review and editing.

### Funding

This research was funded by the French National Research Agency grant ANR-10-IAHU-04 (**E.V**.), the Leducq Foundation grant 16 CVD 02 (**N.T**. and **E.V**.) and the National Institutes

of Health National Heart, Lung, and Blood Institute grant T32 HL007024 (**K.Z**. and **N.T**.).

### Acknowledgements

This work was granted access to the high-performance computing resources of Très Grand Centre de Calcul du CEA (TGCC) under the allocation 2024-A0180310517 madeby Grand Équipement National de Calcul Intensif (GENCI).

Open access publication funding provided by COUPERIN CY26.

### Keywords

adrenergic, electrophysiology, model, simulation, sympathetic, tachycardia, ventricle

## Supporting information

Additional supporting information can be found online in the Supporting Information section at the end of the HTML view of the article. Supporting information files available:

**Peer Review History**
**Video S1**
**Video S2**

