## [Peer Review History · The Journal of Physiology]

A model of beta-adrenergic stimulation in human ventricular cells for tissue-scale simulations of sympathetically-modulated tachycardias

Kelly Zhang, Karl Magtibay, Natalia A Trayanova, and Edward Vigmond

DOI: 10.1113/JP289340

Corresponding author(s): Karl Magtibay (karl.magtibay@ieee.org)

The following individual(s) involved in review of this submission have agreed to reveal their identity: Trine Krogh-Madsen (Referee #1)

Review Timeline:

Submission Date:	27-May-2025
Editorial Decision:	18-Jun-2025
Revision Received:	28-Oct-2025
Editorial Decision:	26-Nov-2025
Revision Received:	08-Dec-2025
Editorial Decision:	06-Jan-2026
Revision Received:	08-Jan-2026
Accepted:	13-Jan-2026

Senior Editor: Bjorn Knollmann

Reviewing Editor: Eleonora Grandi

Transaction Report:

Dear Dr Magtibay,

Re: JP-RP-2025-289340 "A model of beta-adrenergic stimulation in human ventricular cells for tissue-scale simulations of sympathetically-modulated tachycardias" by Kelly Zhang, Karl Magtibay, Natalia A Trayanova, and Edward Vigmond

Thank you for submitting your manuscript to The Journal of Physiology. It has been assessed by a Reviewing Editor and by 2 expert referees and we are pleased to tell you that it is potentially acceptable for publication following satisfactory major revision.

REVISION CHECKLIST:

Please upload two versions of your manuscript text: one with all relevant changes highlighted and one clean version with no

changes tracked. The manuscript file should include all tables and figure legends, but each figure/graph should be uploaded as separate, high-resolution files.

We look forward to receiving your revised submission.

Yours sincerely,

Bjorn Knollmann
Senior Editor
The Journal of Physiology

REQUIRED ITEMS

- Author photo and profile. First or joint first authors are asked to provide a short biography (no more than 100 words for one author or 150 words in total for joint first authors) and a portrait photograph. These should be uploaded and clearly labelled together in a Word document with the revised version of the manuscript. See Information for Authors for further details.

- The Journal of Physiology funds authors of provisionally accepted papers to use the premium BioRender site to create high resolution schematic figures. Follow this link and enter your details and the manuscript number to create and download figures. Upload these as the figure files for your revised submission. If you choose not to take up this offer, we require figures to be of similar quality and resolution. If you are opting out of this service to authors, state this in the Comments section on the Detailed Information page of the submission form. The link provided should only be used for the purposes of this submission. Authors will be charged for figures created on this premium BioRender account if they are not related to this manuscript submission.

- Please upload separate high-quality figure files via the submission form.

- Please ensure that any tables are editable and in Word format, and wherever possible, embedded in the article file itself.

- Please ensure that the Article File you upload is a Word file.

- Your paper contains Supporting Information of a type that we no longer publish, including supplementary tables and figures. Any information essential to an understanding of the paper must be included as part of the main manuscript and figures. The only Supporting Information that we publish are video and audio, 3D structures, program codes and large data files. Your revised paper will be returned to you if it does not adhere to our Supporting Information Guidelines.

- Papers must comply with the Statistics Policy: https://jp.msubmit.net/cgi-bin/main.plex?form_type=display_requirements#statistics.

In summary:

- If n {less than or equal to} 30, all data points must be plotted in the figure in a way that reveals their range and distribution. A bar graph with data points overlaid, a box and whisker plot or a violin plot (preferably with data points included) are

acceptable formats.

- If $n > 30$, then the entire raw dataset must be made available either as supporting information, or hosted on a not-for-profit repository, e.g. FigShare, with access details provided in the manuscript.
- 'n' clearly defined (e.g. x cells from y slices in z animals) in the Methods. Authors should be mindful of pseudoreplication.
- All relevant 'n' values must be clearly stated in the main text, figures and tables.
- The most appropriate summary statistic (e.g. mean or median and standard deviation) must be used. Standard Error of the Mean (SEM) alone is not permitted.
- Exact p values must be stated. Authors must not use 'greater than' or 'less than'. Exact p values must be stated to three significant figures even when 'no statistical significance' is claimed.

- Please include an Abstract Figure file, as well as the Figure Legend text within the main article file. The Abstract Figure is a piece of artwork designed to give readers an immediate understanding of the research and should summarise the main conclusions. If possible, the image should be easily 'readable' from left to right or top to bottom. It should show the physiological relevance of the manuscript so readers can assess the importance and content of its findings. Abstract Figures should not merely recapitulate other figures in the manuscript. Please try to keep the diagram as simple as possible and without superfluous information that may distract from the main conclusion(s). Abstract Figures must be provided by authors no later than the revised manuscript stage and should be uploaded as a separate file during online submission labelled as File Type 'Abstract Figure'. Please also ensure that you include the figure legend in the main article file. All Abstract Figures should be created using BioRender. Authors should use The Journal's premium BioRender account to export high-resolution images. Details on how to use and access the premium account are included as part of this email.

EDITOR COMMENTS

Reviewing Editor:

Both reviewers see potential value in the proposed computational framework, but both note that substantial revision is required. A satisfactory revision should provide clearer rationale and justification for the modeling choices (electrophysiological modeling framework, signaling equations and parameters), more in depth validation against experimental data, and demonstration of added value with respect to existing modeling frameworks.

Senior Editor:

I concur with the reviewing editor's assessment.

REFEREE COMMENTS

Referee #1:

This manuscript by Zhang, Magtibay, et al. first presents a mathematical model of beta-adrenergic stimulation (BARS) in human ventricular myocytes. Such models have been developed previously but this model is less mathematically detailed and therefore more readily allows for tissue level simulations of cardiac arrhythmia-related dynamics. Indeed, the authors carry out such simulations, exploring effects of tissue-level heterogeneity in BARS.

The proposed model does fill a gap in existing model modalities but as detailed below I find the model development lacking in description and rationales. It would also increase impact if the simulated spiral wave dynamics were described in more detail.

Major concerns/comments:

1) The BARS model is based on the Ten Tusscher model of human ventricular cells which is much simpler and therefore more easily simulated at the tissue level. However, in this model repolarization relies much more on IKs than on IKr, in

contrast to other human models, see e.g., Mann et al, doi: 10.1016/j.yjmcc.2016.09.011. For example, Figure S3 shows how block of IKr in the new model has negligible impact on the action potential duration. The authors should discuss this limitation.

2) As a corollary, the reliance on IKs for repolarization in the Ten Tusscher presumably explains why the action potential duration shortens so dramatically with BARS (which increases IKs). Is this amount of shortening (from 300 to 200 ms, at the same pacing rate) physiological?

3) Descriptions of the BARS model development are lacking in detail. As one example, for the scaling of GNa by a factor of 1.9, two references are given but it's not clear if these two references are in agreement on this particular number or (more probable) give two different scalings that potentially surround 1.9. It would be helpful to include a table in the supplement with the experimental data underlying all the BARS-induced changes, including details on the species used in the experiments.

4) It seems that the increased release of Ca from the SR that occurs with BARS is modeled as a scaling of this current although physiologically phosphorylation of the RYRs accelerates release. What is the rationale for modeling increased RYR release as a simple flux scaling?

5) It seems that the authors increase IK1 dramatically with BARS (Fig S1). Is this physiological and if so what is the exact rationale? The explanation provided in the figure caption doesn't quite make sense to me: "To maintain the balance of the potassium current contributions to total repolarization reserve, the IKs curve was used to inform phosphorylation levels of IK1 and IpK." Don't we expect IKs to become more dominant with BARS and therefore the balance to not be maintained? In the main text it is stated that "we finetuned the overall action potential (AP) behaviour to reflect experimental data [40, 31] by scaling the total currents of plateau K⁺ and inward rectifier K⁺ by 3.62 and 2.5, respectively". I strongly disagree with referring to a scaling of IK1 2.5 as fine-tuning. This scaling of IK1 actually is quite critical as it can also be important for conduction velocity and rotation speeds in tissue simulations. Indeed, work from e.g., Jalife's group (ref 23 and others), has shown that the increase in IK1 associated with remodeling in atrial fibrillation induces smaller and faster reentrant waves.

6) For the tissue simulations, it seems that one "cell" is 400² microns² and that this is the spatial resolution of the BARS heterogeneity. Is this physiological? A discussion of this should be included.

7) The finding that the spiral tip meanders to areas with longer action potential duration (fig 8A) is somewhat surprising. Is that a robust finding? What could be a possible explanation?

8) In the conclusions it is stated that "We created a novel ionic model of the ventricular myocyte to incorporate the macroscopic effects of the complex intracellular dynamics due to sympathetic hyperactivity exhibited by MI and HF patients." The "hyperactivity" part comes as a surprise. Is 1 microM ISO hyperactivity? If this is indeed a model of these disease states, shouldn't the model include the remodeling processes associated with them? This should be clearly stated.

Minor concerns/comments:

1) Some of the BARS-induced modifications are given with several significant decimals, e.g., the modification of the KmNa parameter in the NaK pump equation is set to 0.71, and the tissue conductivities have 4 significant decimals. Is this level of accuracy necessary? Is there a rationale for it?

2) The authors state that "We tuned the intracellular parameters of our novel ionic model to reach a steady-state using a two-dimensional (2D) substrate" (page 3). This is not clear to me. What parameters? Is fine-tuning necessary to reach a steady state?

3) Fig 5: The CaT amplitude is increased already at an ISO concentration of 0.01 microM where ICaL and RyR effects are minimal (according to fig S1). It is not clear how that happens. Is the increased CaT mediated by increased Iup (maybe SR loading?) or maybe the increased INaK impacting intracellular Na and Ca concentrations?

4) Fig 7 and text: the authors refer to "heterogeneity (%)" when I believe they mean percentage of virtual cells with BARS which is not synonymous with heterogeneity. For example, in Fig. 7B, if % BARS cells increased beyond 50%, heterogeneity will decrease but one wouldn't expect angular speed to start dropping.

5) It appears the legend for fig 8 is missing. There is just a title but no explanation of the different colors, etc.

6) What type of sigmoid function was used for the fits in fig S1 and why are the fits so asymmetric (i.e., very steep at the high ISO levels)? Is it a fit of the alpha parameter in eqn 1? This should be stated clearly.

Referee #2:

The authors present a single-cell model of β -adrenergic stimulation in human ventricular myocytes, tailored for integration into tissue-scale simulations of sympathetically modulated tachycardias. This work is very important, as spatial heterogeneities in sympathetic activity-particularly in the context of heart failure or post-infarct remodeling-are known contributors to arrhythmogenesis. However, the mechanisms by which these heterogeneities contribute to arrhythmias remain incompletely understood. Computational modeling offers a valuable approach to probe these complex interactions. The study demonstrates that regions of heterogeneous sympathetic stimulation (> 20%) can support stable and rapid rotor formation. These insights can help advance our understanding of how spatially varying autonomic modulation contributes to arrhythmia maintenance and complexity.

While the tissue-level simulations are extremely important and shed light on plausible arrhythmogenic mechanisms under heterogeneous sympathetic stimulation, I have several questions and concerns regarding the development and implementation of the single-cell model:

1. The authors propose using the Ten-Tusscher model for electrophysiology (EP) in combination with a simplified signaling model. They justify this approach by noting that incorporating multiple sets of ODEs-accounting for both EP and signaling-at each node can be computationally intensive, and that their simplified model is better suited for large-scale tissue-level simulations. One could argue that it may not be necessary to solve the signaling dynamics at every node in the tissue. From my understanding, the electrophysiology and signaling ODEs are solved independently at each time point, with phosphorylation information from the signaling model passed to the electrophysiology model. In such a framework, a single-cell simulation (which typically reaches steady state in under five minutes) can be run for the desired stimulus (ISO concentration) to generate a time course of PKA phosphorylation for each target. This time vector can then be used as input for the tissue-level electrophysiology model. At each time step and at every node, the EP ODEs can be solved using the precomputed phosphorylation values (from single cell simulation), effectively eliminating the need to solve the signaling ODEs during the tissue simulation. This approach is not computationally intensive and allows flexibility in choosing any electrophysiological model for the tissue. If the signaling and electrophysiology models are not independent in your framework, I would appreciate clarification or justification for the chosen implementation.

2. Could you please clarify how the proposed signaling model differs from existing detailed β -adrenergic signaling models developed in rabbit, mouse, and canine systems (e.g., (Saucerman et al., 2003; Heijman et al., 2011; Morotti et al., 2021)? Specifically, how simplified is your model in comparison to the Heijman et al. formulation, which includes 57 ODEs? It would be helpful if you could provide details of the updated equations and parameters used in the signaling module-perhaps in the Appendix-so that readers can better understand the extent of simplification. Additionally, does your approach rely solely on fitting the time course of phosphorylated vs. non-phosphorylated targets to experimental data, without explicitly including signaling ODEs? Clarification on this point would be appreciated.

3. Some of the PKA phosphorylation targets included in the proposed model (INa, ICaL, IK1, IKs, IpK, INaK, Irel, Iup, and Ixfer)-do not fully align with those incorporated in existing models that couple β -adrenergic signaling to human electrophysiology, such as the integration of the canine signaling framework (Heijman et al., 2011) with the O'Hara-Rudy model in (Gong et al., 2020) the ToR-ORd model (Doste & Bueno-Orovio, 2021) and the T-world model (Tomek et al., 2025). For instance, (Gong et al., 2020) included PKA effects of INa, ICaL, IKs, Jrel, INaK, Iup, IKb, and Tnl. It would be helpful if you could justify the differences in selected PKA targets across models. This variability makes it challenging for experimentalists and modelers to reconcile and compare results across different frameworks, and a brief rationale for the choices made would greatly improve clarity.

4. This also raises concerns regarding the choice of the Ten-Tusscher electrophysiology model, which-while computationally efficient and commonly used for tissue-level simulations-is relatively dated. More recent human ventricular models such as O'Hara-Rudy 2011 (used in Gong et al., 2021), ToR-ORd 2019 (Doste et al., 2021), and T-world 2025 (Tomek et al., 2025) offer improved fidelity to experimental data and incorporate more physiologically detailed representations of ventricular electrophysiology. Given this, it would be important to evaluate how the proposed signaling model, when coupled with the Ten-Tusscher EP framework, performs in comparison to prior studies that integrated β -adrenergic signaling into more modern electrophysiological models (e.g., Gong 2021, ToR-ORd, Tomek 2025). Are there any improvements in fidelity to experimental data or novel insights yielded by this new implementation? Please clearly justify the motivation for developing a new signaling model in light of existing frameworks and elaborate on its advantages-whether in terms of computational

efficiency, biological accuracy, or predictive capability. If the primary benefit lies in enabling large-scale tissue simulations, please clearly justify against point No.1.

5. Previous signaling models, such as (Heijman et al., 2011) do not include PKA-mediated modulation of IK1. Are there specific experimental references in canine myocytes that support PKA phosphorylation effects on IK1? For instance, (Reilly et al., 2020) reported a ~20% increase in IK1 in response to ISO in mice, but species differences should be considered when generalizing these effects. In the current model, a 150% increase in IK1 due to PKA phosphorylation appears quite high. Such a substantial enhancement can significantly reduce the resting membrane potential, which in turn influences cellular excitability. I haven't come across experimental studies that examine the effects of ISO on resting membrane potential in cardiac cells. Could you please justify the magnitude of IK1 modulation implemented in your model and its physiological plausibility in the context of existing experimental evidence?

6. Are the results presented in Figures 3 and 5 (panels A and B) obtained at a pacing rate of 1 Hz? If so, could you please comment on how these results compare at other pacing frequencies, especially at faster rates under sympathetic stimulation? It would be valuable to understand how the model predictions align with experimental observations across a range of pacing conditions.

7. Please provide details on the steady-state criteria used in the simulations. Specifically, how many beats at each ISO concentration are required for the ionic concentrations and APD to stabilize? For reference, the Gong et al. (2020) model typically requires several hundred beats to reach steady state. Including this information would help clarify the simulation protocol and model behavior.

8. Reference 40 (Priori & Corr, 1990) demonstrated that in canine myocytes exposed to ISO for 10 minutes, intermediate ISO concentrations caused slight prolongation of APD, whereas higher concentrations led to APD shortening. Could you please clarify whether the proposed model is able to replicate this behavior? Including a comparison with this experimental observation would strengthen the model's validation.

9. At very low ISO concentrations (~0.01 μM), the model shows a slight decrease in conduction velocity compared to baseline in some cases. Intuitively, one would expect conduction to increase with any degree of sympathetic stimulation. Could you please clarify the reason for this observation? Additionally, it may be important to consider the effects of ISO on gap junctional coupling (diffusion coefficients) in the tissue simulations, alongside ion channel modulation. Previous studies (e.g., (Xia et al., 2009; Campbell et al., 2014) have demonstrated that β -adrenergic stimulation can influence gap junction conductance, which could significantly impact conduction velocity. Accounting for this could enhance the physiological relevance of the tissue-level model.

10. The tissue-level simulations utilize very high ISO concentrations-1 μM -distributed in different amounts across the tissue. Is such a saturating ISO level necessary for the model to generate fast and stable rotors? It would be helpful to understand how the model behaves at more physiological catecholamine concentrations and lower ISO levels in the range of 0.01 to 0.1 μM . Could you please comment on the stability and dynamics of rotors under these conditions?

11. Are the observed rotor dynamics and stability in the tissue simulations primarily driven by APD heterogeneity? For instance, if similar spatial APD heterogeneity were introduced by varying repolarizing conductances such as gKr or gKs (rather than through ISO clustering), how would the resulting rotor behavior compare? Given that ISO affects multiple targets-including Ca^{2+} handling, conduction velocity, APD, and restitution properties-it would be important to clarify whether the rotor dynamics observed are uniquely attributable to this multi-target modulation, as opposed to simpler models of APD heterogeneity (e.g., via gKr variation alone). A comparative analysis would help distinguish the specific role of sympathetic signaling in arrhythmia mechanisms.

References used in reviewer response:

1. Campbell AS, Johnstone SR, Baillie GS & Smith G (2014). β -Adrenergic modulation of myocardial conduction velocity: Connexins vs. sodium current. *J Mol Cell Cardiol* 77, 147-154.
2. Doste R & Bueno-Orovio A (2021). Multiscale Modelling of β -Adrenergic Stimulation in Cardiac Electromechanical Function. *Mathematics* 9, 1785.
3. Gong JQX, Susilo ME, Sher A, Musante CJ & Sobie EA (2020). Quantitative analysis of variability in an integrated model of human ventricular electrophysiology and β -adrenergic signaling. *J Mol Cell Cardiol* 143, 96-106.
4. Heijman J, Volders PGA, Westra RL & Rudy Y (2011). Local control of β -adrenergic stimulation: Effects on ventricular myocyte electrophysiology and Ca^{2+} -transient. *J Mol Cell Cardiol* 50, 863-871.
5. Morotti S, Liu C, Hegyi B, Ni H, Fogli Iseppe A, Wang L, Pritoni M, Ripplinger CM, Bers DM, Edwards AG & Grandi E (2021). Quantitative cross-species translators of cardiac myocyte electrophysiology: Model training, experimental validation, and applications. *Sci Adv* 7, eabg0927.
6. Priori SG & Corr PB (1990). Mechanisms underlying early and delayed afterdepolarizations induced by catecholamines. *Am J Physiol* 258, H1796-1805.
7. Reilly L, Alvarado FJ, Lang D, Abozeid S, Van Ert H, Spellman C, Warden J, Makielski JC, Glukhov AV & Eckhardt LL (2020). Genetic Loss of IK1 Causes Adrenergic-induced Phase 3 Early Afterdepolarizations and Polymorphic and Bi-directional Ventricular Tachycardia. *Circ Arrhythm Electrophysiol* 13, e008638.
8. Saucerman JJ, Brunton LL, Michailova AP & McCulloch AD (2003). Modeling β -Adrenergic Control of Cardiac Myocyte Contractility in Silico*. *J Biol Chem* 278, 47997-48003.
9. Tomek J, Zhou X, Martinez-Navarro H, Holmes M, Bury T, Berg LA, Tomkova M, Jo E, Nagy N, Bertrand A, Bueno-Orovio A, Colman M, Rodriguez B, Bers D & Heijman J (2025). T-World: A highly general computational model of a human ventricular myocyte. 2025.03.24.645031. Available at: <https://www.biorxiv.org/content/10.1101/2025.03.24.645031v2> [Accessed June 8, 2025].
10. Xia Y, Gong K, Xu M, Zhang Y, Guo J, Song Y & Zhang P (2009). Regulation of gap-junction protein connexin 43 by β -adrenergic receptor stimulation in rat cardiomyocytes. *Acta Pharmacol Sin* 30, 928-934.

END OF COMMENTS

The authors present a single-cell model of β -adrenergic stimulation in human ventricular myocytes, tailored for integration into tissue-scale simulations of sympathetically modulated tachycardias. This work is very important, as spatial heterogeneities in sympathetic activity—particularly in the context of heart failure or post-infarct remodeling—are known contributors to arrhythmogenesis. However, the mechanisms by which these heterogeneities contribute to arrhythmias remain incompletely understood. Computational modeling offers a valuable approach to probe these complex interactions. The study demonstrates that regions of heterogeneous sympathetic stimulation ($> 20\%$) can support stable and rapid rotor formation. These insights can help advance our understanding of how spatially varying autonomic modulation contributes to arrhythmia maintenance and complexity.

While the tissue-level simulations are extremely important and shed light on plausible arrhythmogenic mechanisms under heterogeneous sympathetic stimulation, I have several questions and concerns regarding the development and implementation of the single-cell model:

1. The authors propose using the Ten-Tusscher model for electrophysiology (EP) in combination with a simplified signaling model. They justify this approach by noting that incorporating multiple sets of ODEs—accounting for both EP and signaling—at each node can be computationally intensive, and that their simplified model is better suited for large-scale tissue-level simulations. One could argue that it may not be necessary to solve the signaling dynamics at every node in the tissue. From my understanding, the electrophysiology and signaling ODEs are solved independently at each time point, with phosphorylation information from the signaling model passed to the electrophysiology model. In such a framework, a single-cell simulation (which typically reaches steady state in under five minutes) can be run for the desired stimulus (ISO concentration) to generate a time course of PKA phosphorylation for each target. This time vector can then be used as input for the tissue-level electrophysiology model. At each time step and at every node, the EP ODEs can be solved using the precomputed phosphorylation values (from single cell simulation), effectively eliminating the need to solve the signaling ODEs during the tissue simulation. This approach is not computationally intensive and allows flexibility in choosing any electrophysiological model for the tissue. If the signaling and electrophysiology models are not independent in your framework, I would appreciate clarification or justification for the chosen implementation.

2. Could you please clarify how the proposed signaling model differs from existing detailed β -adrenergic signaling models developed in rabbit, mouse, and canine systems (e.g.,

(Saucerman *et al.*, 2003; Heijman *et al.*, 2011; Morotti *et al.*, 2021)? Specifically, how simplified is your model in comparison to the Heijman *et al.* formulation, which includes 57 ODEs? It would be helpful if you could provide details of the updated equations and parameters used in the signaling module—perhaps in the Appendix—so that readers can better understand the extent of simplification. Additionally, does your approach rely solely on fitting the time course of phosphorylated vs. non-phosphorylated targets to experimental data, without explicitly including signaling ODEs? Clarification on this point would be appreciated.

3. Some of the PKA phosphorylation targets included in the proposed model (INa, ICaL, IK1, IKs, IpK, INaK, Irel, Iup, and Ixfer)—do not fully align with those incorporated in existing models that couple β -adrenergic signaling to human electrophysiology, such as the integration of the canine signaling framework (Heijman *et al.*, 2011) with the O'Hara-Rudy model in (Gong *et al.*, 2020) the ToR-ORd model (Doste & Bueno-Orovio, 2021) and the T-world model (Tomek *et al.*, 2025). For instance, (Gong *et al.*, 2020) included PKA effects of INa, ICaL, IKs, Irel, INaK, Iup, IKb, and Tnl. It would be helpful if you could justify the differences in selected PKA targets across models. This variability makes it challenging for experimentalists and modelers to reconcile and compare results across different frameworks, and a brief rationale for the choices made would greatly improve clarity.

4. This also raises concerns regarding the choice of the Ten-Tusscher electrophysiology model, which—while computationally efficient and commonly used for tissue-level simulations—is relatively dated. More recent human ventricular models such as O'Hara-Rudy 2011 (used in Gong *et al.*, 2021), ToR-ORd 2019 (Doste *et al.*, 2021), and T-world 2025 (Tomek *et al.*, 2025) offer improved fidelity to experimental data and incorporate more physiologically detailed representations of ventricular electrophysiology. Given this, it would be important to evaluate how the proposed signaling model, when coupled with the Ten-Tusscher EP framework, performs in comparison to prior studies that integrated β -adrenergic signaling into more modern electrophysiological models (e.g., Gong 2021, ToR-ORd, Tomek 2025). Are there any improvements in fidelity to experimental data or novel insights yielded by this new implementation? Please clearly justify the motivation for developing a new signaling model in light of existing frameworks and elaborate on its advantages—whether in terms of computational efficiency, biological accuracy, or predictive capability. If the primary benefit lies in enabling large-scale tissue simulations, please clearly justify against point No.1.

5. Previous signaling models, such as (Heijman *et al.*, 2011) do not include PKA-mediated modulation of IK1. Are there specific experimental references in canine myocytes that support PKA phosphorylation effects on IK1? For instance, (Reilly *et al.*, 2020) reported a ~20% increase in IK1 in response to ISO in mice, but species differences should be considered when generalizing these effects. In the current model, a 150% increase in IK1 due to PKA phosphorylation appears quite high. Such a substantial enhancement can significantly reduce the resting membrane potential, which in turn influences cellular excitability. I haven't come across experimental studies that examine the effects of ISO on resting membrane potential in cardiac cells. Could you please justify the magnitude of IK1 modulation implemented in your model and its physiological plausibility in the context of existing experimental evidence?

6. Are the results presented in Figures 3 and 5 (panels A and B) obtained at a pacing rate of 1 Hz? If so, could you please comment on how these results compare at other pacing frequencies, especially at faster rates under sympathetic stimulation? It would be valuable to understand how the model predictions align with experimental observations across a range of pacing conditions.

7. Please provide details on the steady-state criteria used in the simulations. Specifically, how many beats at each ISO concentration are required for the ionic concentrations and APD to stabilize? For reference, the Gong *et al.* (2020) model typically requires several hundred beats to reach steady state. Including this information would help clarify the simulation protocol and model behavior.

8. Reference 40 (Priori & Corr, 1990) demonstrated that in canine myocytes exposed to ISO for 10 minutes, intermediate ISO concentrations caused slight prolongation of APD, whereas higher concentrations led to APD shortening. Could you please clarify whether the proposed model is able to replicate this behavior? Including a comparison with this experimental observation would strengthen the model's validation.

9. At very low ISO concentrations ($\sim 0.01 \mu\text{M}$), the model shows a slight decrease in conduction velocity compared to baseline in some cases. Intuitively, one would expect conduction to increase with any degree of sympathetic stimulation. Could you please clarify the reason for this observation? Additionally, it may be important to consider the effects of ISO on gap junctional coupling (diffusion coefficients) in the tissue simulations, alongside ion channel modulation. Previous studies (e.g., (Xia *et al.*, 2009; Campbell *et al.*, 2014) have demonstrated that β -adrenergic stimulation can influence gap junction conductance, which could significantly impact conduction velocity. Accounting for this could enhance the physiological relevance of the tissue-level model.

10. The tissue-level simulations utilize very high ISO concentrations— $1 \mu\text{M}$ —distributed in different amounts across the tissue. Is such a saturating ISO level necessary for the model to generate fast and stable rotors? It would be helpful to understand how the model behaves at more physiological catecholamine concentrations and lower ISO levels in the range of 0.01 to $0.1 \mu\text{M}$. Could you please comment on the stability and dynamics of rotors under these conditions?

11. Are the observed rotor dynamics and stability in the tissue simulations primarily driven by APD heterogeneity? For instance, if similar spatial APD heterogeneity were introduced by varying repolarizing conductances such as g_{Kr} or g_{Ks} (rather than through ISO clustering), how would the resulting rotor behavior compare? Given that ISO affects multiple targets—including Ca^{2+} handling, conduction velocity, APD, and restitution properties—it would be important to clarify whether the rotor dynamics observed are uniquely attributable to this multi-target modulation, as opposed to simpler models of APD heterogeneity (e.g., via g_{Kr} variation alone). A comparative analysis would help distinguish the specific role of sympathetic signaling in arrhythmia mechanisms.

References:

1. Campbell AS, Johnstone SR, Baillie GS & Smith G (2014). β -Adrenergic modulation of myocardial conduction velocity: Connexins vs. sodium current. *J Mol Cell Cardiol* **77**, 147–154.
2. Doste R & Bueno-Orovio A (2021). Multiscale Modelling of β -Adrenergic Stimulation in Cardiac Electromechanical Function. *Mathematics* **9**, 1785.
3. Gong JQX, Susilo ME, Sher A, Musante CJ & Sobie EA (2020). Quantitative analysis of variability in an integrated model of human ventricular electrophysiology and β -adrenergic signaling. *J Mol Cell Cardiol* **143**, 96–106.
4. Heijman J, Volders PGA, Westra RL & Rudy Y (2011). Local control of β -adrenergic stimulation: Effects on ventricular myocyte electrophysiology and Ca^{2+} -transient. *J Mol Cell Cardiol* **50**, 863–871.
5. Morotti S, Liu C, Hegyi B, Ni H, Fogli Iseppe A, Wang L, Pritoni M, Ripplinger CM, Bers DM, Edwards AG & Grandi E (2021). Quantitative cross-species translators of cardiac myocyte electrophysiology: Model training, experimental validation, and applications. *Sci Adv* **7**, eabg0927.
6. Priori SG & Corr PB (1990). Mechanisms underlying early and delayed afterdepolarizations induced by catecholamines. *Am J Physiol* **258**, H1796-1805.
7. Reilly L, Alvarado FJ, Lang D, Abozeid S, Van Ert H, Spellman C, Warden J, Makielski JC, Glukhov AV & Eckhardt LL (2020). Genetic Loss of IK1 Causes Adrenergic-induced Phase 3 Early Afterdepolarizations and Polymorphic and Bi-directional Ventricular Tachycardia. *Circ Arrhythm Electrophysiol* **13**, e008638.
8. Saucerman JJ, Brunton LL, Michailova AP & McCulloch AD (2003). Modeling β -Adrenergic Control of Cardiac Myocyte Contractility in Silico*. *J Biol Chem* **278**, 47997–48003.
9. Tomek J, Zhou X, Martinez-Navarro H, Holmes M, Bury T, Berg LA, Tomkova M, Jo E, Nagy N, Bertrand A, Bueno-Orovio A, Colman M, Rodriguez B, Bers D & Heijman J (2025). T-World: A highly general computational model of a human ventricular myocyte. 2025.03.24.645031. Available at: <https://www.biorxiv.org/content/10.1101/2025.03.24.645031v2> [Accessed June 8, 2025].
10. Xia Y, Gong K, Xu M, Zhang Y, Guo J, Song Y & Zhang P (2009). Regulation of gap-junction protein connexin 43 by β -adrenergic receptor stimulation in rat cardiomyocytes. *Acta Pharmacol Sin* **30**, 928–934.

Title: A model of beta-adrenergic stimulation in human ventricular cells for tissue-scale simulations of sympathetically-modulated tachycardias

Authors: Kelly Zhang*, Karl Magtibay*, Natalia Trayanova, and Edward Vigmond (*shared first authors)

Manuscript ID: JP-RP-2025-289340

Response to Reviewers

The following sections contain our responses to each comment the reviewers provided.

Original reviewer comments are **bolded in blue**, and our responses are in black.

We wrote in *italics* the appropriate sections where the changes in our revised manuscript could be found. The [...] notation indicates continuation of paragraphs or sentences that may be unrelated to the reviewer's queries.

EDITOR COMMENTS

Reviewing Editor:

Both reviewers see potential value in the proposed computational framework, but both note that substantial revision is required. A satisfactory revision should provide clearer rationale and justification for the modeling choices (electrophysiological modeling framework, signaling equations and parameters), more in depth validation against experimental data, and demonstration of added value with respect to existing modeling frameworks.

Senior Editor:

I concur with the reviewing editor's assessment.

Author response:

We thank the reviewers and the editors for allowing us to strengthen our work on modelling the beta-adrenergic stimulation (BARS) effects on ventricular tachycardias (VT). Below, we address the comments and concerns of each referee, where applicable and relevant to our study, and cite the revisions in our revised manuscript.

REFEREE COMMENTS

Referee #1:

This manuscript by Zhang, Magtibay, et al. first presents a mathematical model of beta-adrenergic stimulation (BARS) in human ventricular myocytes. Such models have been developed previously but this model is less mathematically detailed and therefore more readily allows for tissue level simulations of cardiac arrhythmia-related dynamics.

Indeed, the authors carry out such simulations, exploring effects of tissue-level heterogeneity in BARS.

The proposed model does fill a gap in existing model modalities but as detailed below I find the model development lacking in description and rationales. It would also increase impact if the simulated spiral wave dynamics were described in more detail.

Author response:

We thank the reviewer for their valuable critique of our work. Please find below our detailed responses to each of their comments and concerns.

Major concerns/comments:

1) The BARS model is based on the Ten Tusscher model of human ventricular cells which is much simpler and therefore more easily simulated at the tissue level. However, in this model repolarization relies much more on IKs than on IKr, in contrast to other human models, see e.g., Mann et al, doi: 10.1016/j.yjmcc.2016.09.011. For example, Figure S3 shows how block of IKr in the new model has negligible impact on the action potential duration. The authors should discuss this limitation.

Author response:

We agree that the ten Tusscher model formulation places a greater reliance on IKs than on IKr for ventricular repolarization, which differs from other human ventricular cell models and experimental data in which the two currents are more balanced. This limitation is inherited from the original ten Tusscher framework and is not unique to our implementation. As presented in the revised *Methods* and *Results*, we upregulated IKs to a lesser degree and modified its gating kinetics to be consistent with other BARS implementations. While this upregulation may still accentuate the discrepancy in IKs/IKr balance, our model now reproduces a more physiological degree of APD shortening and rate dependence. Since IKs plays the major role in APD shortening during BARS (Volders 2003), the imbalance between IKr and IKs likely has a smaller influence on the BARS-specific results we present. Nevertheless, we acknowledge that the baseline model's inherent limitation may become more significant when considering concurrent effects of other signaling pathways, such as parasympathetic stimulation, which is known to affect IKr (Koncz 2022).

Methods

[...]

We increased the maximal conductance G_{Ks} by a factor of 2.5, shifted the activation gate by -5 mV, and increased the time of activation for the slow delayed rectifier K^+ channels (Ks)

[...]

Results

[...]

In Fig. 4C, we present a 2.3 times increase in I_{Ks} I-V peak current density for a 3000 ms duration clamp pulse, within the range of experimental data (Volders et al. 2003).

[...]

At 1 μ M ISO the fraction of phosphorylated substrates is maximized, resulting in plateaued CaT and APD at 2.25 and 0.83 times the baseline values, respectively.

[...]

2) As a corollary, the reliance on IKs for repolarization in the Ten Tusscher presumably explains why the action potential duration shortens so dramatically with BARS (which increases IKs). Is this amount of shortening (from 300 to 200 ms, at the same pacing rate) physiological?

Author response:

We thank the reviewers for bringing up this important point. We have revised the model so that the APD shortening falls within a more physiological range. This change has been implemented as described in our response to comment 1 and reflected in the revised Results.

3) Descriptions of the BARS model development are lacking in detail. As one example, for the scaling of G_{Na} by a factor of 1.9, two references are given but it's not clear if these two references are in agreement on this particular number or (more probable) give two different scalings that potentially surround 1.9. It would be helpful to include a table in the supplement with the experimental data underlying all the BARS-induced changes, including details on the species used in the experiments.

Author response:

In the revised manuscript, we indicated in Table 1 of our main manuscript the experimental data underlying all BARS-induced changes, including the species used in each study. We have also revised the main text to provide additional detail on how scaling factors were selected, clarifying whether our chosen values fall within reported experimental ranges or, in cases where exact agreement was not possible, were instead adjusted to ensure that macroscopic behaviors (e.g. APD, CV) remained consistent with experimental observations.

Results

[...]

We show in Fig.A that increasing the maximal conductance of phosphorylated I_{Na} (G_{Na}) results in a larger current amplitude (-358.2 vs -332.9 pA/pF) and a 1.5-fold increase in peak current density of the I-V curve, above the range reported in Baba et al. [2] (1.32 ± 0.052 fold). Additionally, while inactivation gating is accordingly shifted in our model, no modifications are included for the activation gate as we found that incremental negative shifts in the half-activation value produced exaggerated CV increases. The discrepancies between the model modifications and experimental data were adopted so the resulting CV increase, as presented in section 3.2, remains in line with the estimated impact from I_{Na} alone [4].

[...]

4) It seems that the increased release of Ca from the SR that occurs with BARS is modeled as a scaling of this current although physiologically phosphorylation of the the RYRs accelerate release. What is the rationale for modeling increased RYR release as a simple flux scaling?

Author response:

In the revised model, we have reworked the BARS-induced increase in SR Ca release to include both increased opening and closing rates of the ryanodine receptors, as well as an increase in conductance. This approach is more consistent with the underlying physiology and with other published implementations of BARS effects. Corresponding details have been added to the Methods section.

[...]

We modified the Ca^{2+} handling currents by following the protocol in Gong et al. [18] With $ICaL$ held in its baseline state and intracellular Ca^{2+} held constant, we parameterized the ryanodine receptor current, sarcoplasmic reticulum uptake current, and cytoplasmic Ca^{2+} buffer from their baselines due to phosphorylation. Specifically, we increased the maximum conductance of the ryanodine receptor (G_{rel}) by a factor of 1.6, the channel's opening rate by 1.4, and closing rate by 1.8. To offset the buildup of Ca^{2+} in the sarcolemmal subspace (CaSS) during an AP's plateau phase and prevent spurious CaT, the rate of Ca^{2+} transfer between the subspace and bulk cytosol was also increased by a factor of 2.1, represented as the maximal conductance of G_{xfer} .

[...]

5) It seems that the authors increase IK1 dramatically with BARS (Fig S1). Is this physiological and if so what is the exact rationale? The explanation provided in the the figure caption doesn't quite make sense to me: "To maintain the balance of the potassium current contributions to total repolarization reserve, the IKs curve was used to inform phosphorylation levels of IK1 and IpK." Don't we expect IKs to become more dominant with BARS and therefore the balance to not be maintained? In the main text it is stated that "we finetuned the overall action potential (AP) behaviour to reflect experimental data [40, 31] by scaling the total currents of plateau K^+ and inward rectifier K^+ by 3.62 and 2.5, respectively". I strongly disagree with referring to a scaling of IK1 2.5 as fine-tuning. This scaling of IK1 actually is quite critical as it can also be important for conduction velocity and rotation speeds in tissue simulations. Indeed, work from e.g., Jalife's group (ref 23 and others), has shown that the increase in IK1 associated with remodeling in atrial fibrillation induces smaller and faster reentrant waves.

Author response:

We thank the reviewer for the detailed comment. While in our old model the adjustment of IK1 had minimal effect on resting potential, it affected the steepness of phase 3 in the AP. We recognize that its inclusion was not well justified physiologically and could potentially have important implications for reentrant dynamics especially in full organ simulations. In the revised model, we have therefore removed this adjustment entirely, to maintain consistency with experimental reports of BARS effects and with other published models of β -adrenergic signaling.

6) For the tissue simulations, it seems that one "cell" is 400^2 microns² and that this is

the spatial resolution of the BARS heterogeneity. Is this physiological? A discussion of this should be included.

Author response:

Our simulations treat cardiac tissues as a syncytium, a common practice in tissue-level simulations (Clayton et al. 2011 and Plonsey 1988). Therefore, each mesh element does not correspond to a cell. Although we can create tissue sheets with fine mesh resolutions to match the length of a cardiomyocyte and even finer mesh elements at the locations of beta-adrenergic receptors, each simulation will require more computational resources (i.e., compute time and power), which will negate our purpose of creating a BARS model for tissue-level simulations. Instead, we implemented BARS using spatial proportions to maintain the relative densities of beta-adrenergic receptors identified by Kawano et al. (2003) in human ventricles and still capture the macroscopic effects of BARS on an existing VT. We revised our manuscript to include a brief discussion on the spatial resolution of our BARS and two-dimensional substrates.

Limitations

[...]

We recognize that the spatial resolution of our sheets is coarse, which limits their ability to represent the adrenergic stimulation of a cardiomyocyte accurately. Tissue- and organ-level simulations typically treat cardiac substrates as a syncytium; thus, each mesh element in a model represents a collection of cardiomyocytes, an accepted practice in cardiac tissue modelling studies [5, 36]. Hence, we modelled the distribution of adrenergically-stimulated cardiomyocytes in area proportions to be consistent with histological evidence[24] and to represent their macroscopic effects on existing VTs sufficiently.

[...]

7) The finding that the spiral tip meanders to areas with longer action potential duration (fig 8A) is somewhat surprising. Is that a robust finding? What could be a possible explanation?

Author response:

Ten Tusscher and Panfilov (2002) have also demonstrated that spiral wave tips tend to migrate to areas with longer APDs in the presence of repolarization gradients, however the mechanism of this phenomenon has not been explained. Although this finding was consistent across all substrates using the previous version of our model, we did not observe the same rotor behaviour in our simulations using our revised model. Nonetheless, the primary rotor splinters at the border of the areas with the largest APD and CV difference, however the manner of how they splinter is dependent on the spatial BARS gradient and ISO concentrations.

Below, we provide excerpts from our revised manuscript of our new findings. We also provide a snippet of Fig. 8 from our revised manuscript.

Results - Tissue simulations - Effect of sympathetic spatial gradients on rotors

[...] Specifically, wavelets could be splintered from the primary rotor at the boundary where the APD and CV gradient are the greatest. For BAR-stimulated substrates with 0.01 μM ISO, wavelets began to form only after a rotor had been sustained (Fig. 8A). In contrast, primary rotors are immediately splintered to wavelets in BAR-stimulated substrates with ISO concentrations $>0.1\mu\text{M}$ (Fig. 8B and C). [...]

8) In the conclusions it is stated that "We created a novel ionic model of the ventricular myocyte to incorporate the macroscopic effects of the complex intracellular dynamics due to sympathetic hyperactivity exhibited by MI and HF patients. " The "hyperactivity" part comes as a surprise. Is 1 microM ISO hyperactivity? If this is indeed a model of these disease states, shouldn't the model include the remodeling processes associated with them? This should be clearly stated.

Author response:

We concede that the term “hyperactivity” is an inappropriate descriptor. We aimed to demonstrate an application of our new cell model, specifically to study sympathetically mediated VTs. Heterogeneous sympathetic nerve sprouting in non-diseased regions occurs as a compensatory mechanism for the loss of cardiac function due to MI or HF, creating sympathetically hyperactive regions. More importantly, sprouted sympathetic nerves maintain their adrenergic signalling pathway as demonstrated by Zhu et al. (2022). Similarly, we qualified “hyperactivity” by increasing the BARS spatial density (previously referred to as heterogeneity in our initial submission) in non-diseased substrates, regardless of the ISO concentration. In the same study, Zhu et al. (2022) also demonstrated that sprouted sympathetic nerves at the border of diseased substrates exhibit altered or suppressed adrenergic signalling mechanisms compared to those in non-diseased substrates. At the time of writing, sprouted sympathetic nerves at the border zone have yet to be characterized experimentally; hence, there is insufficient experimental data to create a diseased model and their interaction with cardiac tissues.

In our revised manuscript, we have corrected our terminology to state clearly the aspects of sympathetic remodelling that we simulated. Below are some excerpts from our revised manuscript where corrections can be found.

Introduction

[...] Using our proposed model, we simulated the isolated effects of varying **spatial densities** of sympathetic stimulation and spatial gradients on an existing ventricular tachycardia (VT) within **non-diseased cardiac substrates** with altered distributions of sympathetic elements.

[...]

Methods, Application to tissue-level simulations, Two-dimensional substrates, BARS spatial density, and rotor generation

[...]

We designated random elements of the square sheet to be adrenergically stimulated, simulating **sympathetic remodelling of non-diseased, two-dimensional (2D) cardiac substrates** due to heterogeneously sprouted nerves. [...]

Conclusions

We created a novel ionic model of the ventricular myocyte to incorporate the macroscopic effects of the complex intracellular dynamics due to sympathetic **remodelling** exhibited by MI and HF patients. [...]

Minor concerns/comments:

1) Some of the BARS-induced modifications are given with several significant decimals, e.g., the modification of the KmNa parameter in the NaK pump equation is set to 0.71, and the tissue conductivities have 4 significant decimals. Is this level of accuracy necessary? Is there a rationale for it?

Author response:

This level of accuracy for tissue conductivities is necessary to achieve specific conduction velocities. Conductivities of simulated tissues are iteratively tuned until the calculated conduction velocity along a cable is as close to the desired value. The stopping point for this process is decided whether the conduction velocity within a cable is within a specified range, which in our case is within $\pm 10^{-4}$.

2) The authors state that "We tuned the intracellular parameters of our novel ionic model to reach a steady-state using a two-dimensional (2D) substrate" (page 3). This is not clear to me. What parameters? Is fine-tuning necessary to reach a steady state?

Author response:

This particular statement in our manuscript is incorrect. We used a cell element to tune the intracellular parameters of our single-cell model to a steady state. We then incorporated these steady-state intracellular parameters into a tissue cable to tune the longitudinal and transverse conductivities, allowing us to maintain a 60 cm/s wave in a substrate with unidirectional fibre orientation. Model tuning is a necessary step in simulations to ensure that the ordinary differential equations involved will converge to stable and predictable solutions, such as those under a steady state. In this manner, model tuning enables us to examine the effect of BARS on cardiac electrophysiology, rather than its impact on the model's stability.

In our revised manuscript, we have revised this sentence to state our procedure accurately. Also, below is a table of the relevant single-cell parameters showing their tuned steady-state values.

Methods, Applications to tissue-level simulations, Two-dimensional substrates, BARS spatial density, and rotor generation

We used a cell element to tune the intracellular parameters of our novel ionic model to reach a steady state, stimulated with a 60 $\mu\text{A}/\text{cm}^2$, 0.5 ms transmembrane current, with a 600 ms basic CL over 60 s. Then, we used our ionic model's steady-state parameters to adjust the longitudinal ($\sigma_L = 0.1608 \text{ S/m}$) and transverse ($\sigma_T = 0.0448 \text{ S/m}$) conductivities of a cable to match a physiological ventricular tissue with approximately 60 cm/s CV[13]. [...]

Model parameter	Shorthand	Normal (ISO = 0)	Low dose (ISO = 0.01)	Med dose (ISO = 0.10)	High dose (ISO = 1.00)
Intracellular $[\text{Ca}^{2+}]$	Ca_i	0.000126258	0.000112131	9.67247e-05	9.3757e-05
SR $[\text{Ca}^{2+}]$	Ca_{SR}	4.07161	4.12462	3.59112	3.47083
Subspace $[\text{Ca}^{2+}]$	Ca_{SS}	0.000330036	0.000225711	0.000189988	0.000184108
Vm-dependent activation of baseline ICaL	D	3.43703e-05	3.30172e-05	3.19448e-05	3.18309e-05
Vm-dependent activation of phosphorylated ICaL	D_{pka}	5.04197e-06	4.77333e-06	4.56317e-06	4.54102e-06
Slow Vm-dependent inactivation of baseline ICaL	F	0.827341	0.856499	0.881104	0.887315
Subspace $[\text{Ca}^{2+}]$ -dependent inactivation of baseline ICaL	F_{CaSS}	0.998317	0.999089	0.998948	0.99897
Fast Vm-dependent inactivation of baseline ICaL	F_2	0.986058	0.989966	0.992414	0.993038
Slow Vm-	F_{pka}	0.828196	0.857569	0.882079	0.888417

dependent inactivation of phosphorylated ICaL					
Subspace $[Ca^{2+}]$ -dependent inactivation of phosphorylated ICaL	$F_{CaSSpka}$	1.19718	1.19853	1.19826	1.1983
Fast Vm-dependent inactivation of phosphorylated ICaL	$F2_{pka}$	1.07782	1.08353	1.08708	1.08798
Maximal conductance of ICaL	G_{CaL}	3.98e-05	3.98e-05	3.98e-05	3.98e-05
Maximal conductance of IKr	G_{Kr}	0.153	0.153	0.153	0.153
Maximal conductance of IKs	G_{Ks}	0.392	0.392	0.392	0.392
Maximal conductance of Ito	G_{to}	0.294	0.294	0.294	0.294
Fast Vm-dependent inactivation of baseline INa	H	0.740629	0.748985	0.755685	0.756412
Fast Vm-dependent inactivation of baseline Late INa	HL	0.326194	0.344614	0.354399	0.358184
Fast Vm-dependent inactivation of phosphorylated INa	H_{pka}	0.576124	0.587418	0.596575	0.5976
Slow Vm-dependent inactivation of baseline INa	J	0.713746	0.728616	0.736814	0.739047
Slow Vm-dependent inactivation of phosphorylated INa	J_{pka}	0.551873	0.568226	0.578151	0.580578
Intracellular $[K^+]$	K_i	135.751	135.828	135.087	135.042
Vm-dependent activation of baseline INa	M	0.00176796	0.00165848	0.00157344	0.00156456
Vm-dependent activation of baseline Late INa	ML	0.000327432	0.000309235	0.000295031	0.000293542
Intracellular $[Na^+]$	Na_i	9.15116	9.07022	9.89962	9.96616
Vm-dependent inactivation gate of vaseline Ito	R	2.47694e-08	2.3552e-08	2.25972e-08	2.24951e-08
Resting close state of baseline RyR	R_{-}	0.93742	0.955123	0.953693	0.9542

Resting close stated of phosphorylated RyR	R_pka	1.45643	1.40709	1.40566	1.40544
Vm-dependent activation of baseline Ito	S	0.999972	0.999973	0.999973	0.999973
Vm-dependent activation of baseline IKr	X _{r1}	0.00318887	0.00181741	0.00135947	0.00119472
Vm-dependent inactivation of baseline IKr	X _{r2}	0.469686	0.472815	0.475388	0.475669
Vm-dependent activation of baseline IKs	X _s	0.00717035	0.00586231	0.00506634	0.004858
Vm-dependent activation of phosphorylated IKs	X _{spka}	0.0472726	0.0376077	0.0300908	0.0282241

3) Fig 5: The CaT amplitude is increased already at an ISO concentration of 0.01microM where ICaL and RyR effects are minimal (according to fig S1). It is not clear how that happens. Is the increased CaT mediated by increased Iup (maybe SR loading?) or maybe the increased INaK impacting intracellular Na and Ca concentrations?

Author response:

To investigate the mechanism underlying the CaT increase, we paced the model for 1000 beats at 1Hz under 0.01uM ISO but disabled phosphorylation for each BARS-modified current one at a time. Disabling phosphorylation of SR Ca²⁺ uptake (Iup) produced the largest reductions in CaT amplitude, indicating that enhanced uptake and SR load is a primary driver of CaT increase.

4) Fig 7 and text: the authors refer to "heterogeneity (%)" when I believe they mean percentage of virtual cells with BARS which is not synonymous with heterogeneity. For example, in Fig. 7B, if % BARS cells increased beyond 50%, heterogeneity will decrease but one wouldn't expect angular speed to start dropping.

Author response:

We agree with the reviewer that the term "heterogeneity" does not accurately represent our simulations and results. Thus, in our revised manuscript, we have replaced this term with "spatial density" and "spatial gradient" where applicable.

5) It appears the legend for fig 8 is missing. There is just a title but no explanation of the different colors, etc.

Author response:

The colours were meant to separate the primary rotor and its wavelets. From the previous version of our model, the primary rotor mixed with its wavelets. However, in our revised model the primary rotor and wavelets remain sufficiently distant from each other and so they can be identified visually. So, we revised Fig. 8 to have the primary rotors and wavelets share the same colour. (see Author Response to Major Concern #7)

6) What type of sigmoid function was used for the fits in fig S1 and why are the fits so asymmetric (i.e., very steep at the high ISO levels)? Is it a fit of the alpha parameter in eqn 1? This should be stated clearly.

Author response:

The portion of our framework that describes the effects of beta-adrenergic receptor signalling on EP is a set of logarithmic dose-response functions, relating ISO concentration to the phosphorylation level of a target channel. This phosphorylation level is the alpha parameter in eqn 1. In our revised model, the curves for INa, INaK, IreI, Iup/Ixfer, IKs, and Kbufc are symmetric and represented by 4-parameter curves. IpK and ICal are represented by 5-parameter curves to capture the asymmetric response to higher levels of ISO as seen experimentally and from the Heijman signalling model simulations.

Referee #2:

The authors present a single-cell model of β -adrenergic stimulation in human ventricular myocytes, tailored for integration into tissue-scale simulations of sympathetically modulated tachycardias. This work is very important, as spatial heterogeneities in sympathetic activity-particularly in the context of heart failure or post-infarct remodeling-are known contributors to arrhythmogenesis. However, the mechanisms by which these heterogeneities contribute to arrhythmias remain

incompletely understood. Computational modeling offers a valuable approach to probe these complex interactions. The study demonstrates that regions of heterogeneous sympathetic stimulation ($> 20\%$) can support stable and rapid rotor formation. These insights can help advance our understanding of how spatially varying autonomic modulation contributes to arrhythmia maintenance and complexity.

While the tissue-level simulations are extremely important and shed light on plausible arrhythmogenic mechanisms under heterogeneous sympathetic stimulation, I have several questions and concerns regarding the development and implementation of the single-cell model:

Author response:

We thank the reviewer for their valuable critique of our work. Please find below our detailed responses to each of their comments and concerns.

1. The authors propose using the Ten-Tusscher model for electrophysiology (EP) in combination with a simplified signaling model. They justify this approach by noting that incorporating multiple sets of ODEs-accounting for both EP and signaling-at each node can be computationally intensive, and that their simplified model is better suited for large-scale tissue-level simulations. One could argue that it may not be necessary to solve the signaling dynamics at every node in the tissue. From my understanding, the electrophysiology and signaling ODEs are solved independently at each time point, with phosphorylation information from the signaling model passed to the electrophysiology model. In such a framework, a single-cell simulation (which typically reaches steady state in under five minutes) can be run for the desired stimulus (ISO concentration) to generate a time course of PKA phosphorylation for each target. This time vector can then be used as input for the tissue-level electrophysiology model. At each time step and at every node, the EP ODEs can be solved using the precomputed phosphorylation values (from single cell simulation), effectively eliminating the need to solve the signaling ODEs during the tissue simulation. This approach is not computationally intensive and allows flexibility in choosing any electrophysiological model for the tissue. If the signaling and electrophysiology models are not independent in your framework, I would appreciate clarification or justification for the chosen implementation.

Author response:

We appreciate the reviewer's insight and hope to clarify our implementation. The signalling portion of our framework serves the same purpose as passing phosphorylation levels precomputed from the decoupled Heijman signalling model at various stimuli. The key point is that our dose-response functions provide a continuous mapping between ISO concentration and phosphorylation levels for the target currents, enabling interpolation across a wide range of ISO values. This eliminates the need to rerun a stand-alone signaling model each time phosphorylation information is required for a slightly different ISO level. Our framework aims to provide practical advantages for organ-scale simulations, where incorporating the

phosphorylation levels for heterogeneous innervation regions or denervation gradients would otherwise require a large number of cell-level simulations and considerable setup to assign vectors at each node.

2. Could you please clarify how the proposed signaling model differs from existing detailed β -adrenergic signaling models developed in rabbit, mouse, and canine systems (e.g., (Saucerman et al., 2003; Heijman et al., 2011; Morotti et al., 2021)? Specifically, how simplified is your model in comparison to the Heijman et al. formulation, which includes 57 ODEs? It would be helpful if you could provide details of the updated equations and parameters used in the signaling module—perhaps in the Appendix—so that readers can better understand the extent of simplification. Additionally, does your approach rely solely on fitting the time course of phosphorylated vs. non-phosphorylated targets to experimental data, without explicitly including signaling ODEs? Clarification on this point would be appreciated.

Author response:

The portion of our framework that describes the effects of beta-adrenergic receptor signalling on EP is simply a set of logarithmic dose-response functions, relating ISO concentration to the phosphorylation level of a target channel. This phosphorylation level is the alpha parameter in equation 1. In our revised model, the curves for INa, INaK, Irel, Iup/Ixfer, IKs, and Kbufc are symmetric and represented by 4-parameter curves. IpK and ICaL are represented by 5-parameter curves to capture the asymmetric response to higher levels of ISO as seen experimentally and from the Heijman signalling model simulations.

Our approach does not fit to time course data of phosphorylation. Each function maps ISO concentration to the steady-state phosphorylation level for one of the nine target currents. Our framework focuses on compatibility with organ-level simulations, where sub-minute durations of simulated data can require hours of realtime computation. Such simulations are inherently too short to capture the full temporal dynamics of β -adrenergic signaling, which evolve over the scale of minutes to hours (Khalilimeybodi 2018). Our framework is more geared for examining the effect of spatially heterogeneous stimulation levels—i.e. regions already at different stages of the BARS cascade—on excitation propagation, rather than for capturing the temporal effect of BARS on arrhythmogenesis.

3. Some of the PKA phosphorylation targets included in the proposed model (INa, ICaL, IK1, IKs, IpK, INaK, Irel, Iup, and Ixfer)—do not fully align with those incorporated in existing models that couple β -adrenergic signaling to human electrophysiology, such as the integration of the canine signaling framework (Heijman et al., 2011) with the O'Hara-Rudy model in (Gong et al., 2020) the ToR-ORd model (Doste & Bueno-Orovio, 2021) and the T-world model (Tomek et al., 2025). For instance, (Gong et al., 2020) included PKA effects of INa, ICaL, IKs, Jrel, INaK, Iup, IKb, and TnI. It would be helpful if you could justify the differences in selected PKA targets across models. This variability makes it challenging for experimentalists and

modelers to reconcile and compare results across different frameworks, and a brief rationale for the choices made would greatly improve clarity.

Author response:

We have taken into account the first round of reviews, which provided valuable feedback that helped improve our model. In response to the comments, we have revised our model so that the modifications are more consistent with both experimental observations and previously published implementations of β -adrenergic signaling (e.g., Gong et al.). Our updated model now includes modifications to nine targets—INa, INaK, IKs, IpK, ICaL, Iup, IreI, Ixfer, Kbufc—as summarized in Table 1. Among these, the scaling of IpK serves as a parallel to the background K⁺ currents in other formulations; the scaling of Kbufc serves as a parallel to Troponin I. The only current we modify that is not typically adjusted in previous implementations is Ixfer, for which we include a rationale in the *Methods* section.

[...] To offset the buildup of Ca²⁺ in the sarcolemmal subspace (CaSS) during an AP's plateau phase and prevent spurious CaT, the rate of Ca²⁺ transfer between the subspace and bulk cytosol was also increased by a factor of 2.1, represented as the maximal conductance of G_{xfer}. [...]

4. This also raises concerns regarding the choice of the Ten-Tusscher electrophysiology model, which-while computationally efficient and commonly used for tissue-level simulations-is relatively dated. More recent human ventricular models such as O'Hara-Rudy 2011 (used in Gong et al., 2021), ToR-ORd 2019 (Doste et al., 2021), and T-world 2025 (Tomek et al., 2025) offer improved fidelity to experimental data and incorporate more physiologically detailed representations of ventricular electrophysiology. Given this, it would be important to evaluate how the proposed signaling model, when coupled with the Ten-Tusscher EP framework, performs in comparison to prior studies that integrated β -adrenergic signaling into more modern electrophysiological models (e.g., Gong 2021, ToR-ORd, Tomek 2025). Are there any improvements in fidelity to experimental data or novel insights yielded by this new implementation? Please clearly justify the motivation for developing a new signaling model in light of existing frameworks and elaborate on its advantages-whether in terms of computational efficiency, biological accuracy, or predictive capability. If the primary benefit lies in enabling large-scale tissue simulations, please clearly justify against point No.1.

Author response:

We agree that recent human ventricular models such as O'Hara-Rudy 2011 and ToR-ORd 2019, incorporate significant refinements and improved fidelity to experimental data. Our motivation for coupling the β -adrenergic signaling module with the ten Tusscher electrophysiology framework, however, is not to propose a fundamentally new cell model with improved biophysiological detail. Rather, our focus is on developing a model that enables organ-level simulations and can provide insight into the arrhythmogenic properties of heterogeneously stimulated tissue.

We selected the Ten Tusscher model due to its well-characterized behavior, computational efficiency, and extensive history of integration into multiscale arrhythmia simulations. The baseline ORd model (2011) contains 41 state variables and ToR-ORd (2019) 45, compared to only 19 ODEs in the baseline TT2 model (2006). Recent benchmarking by Dong et al. (2025) highlights several important differences between the TT2 and ORD-based formulations. In isotropic 2D slabs, reentry generated using ToR-ORd was less stable, with occasional wave break or unsustainable activity and higher $[Ca^{2+}]_i$ spatial heterogeneity—particularly at larger time steps. By contrast, TT2 showed minimal variation in rotor morphology and location across a broad range of time steps (5–30 μ s). Moreover, in 3D ventricular simulations, both models reproduced similar VT mechanisms and identified critical conduction channels relevant for ablation, but ToR-ORd occasionally produced non-clinical “pseudo-reentries” linked to numerical instability. Lastly, ToR-ORd required ~ 1.5 x the compute time of TT2 at a 25 μ s time step and ~ 1.7 x at 10 μ s.

Since we aim to use our model to investigate how heterogeneity (e.g. percolated stimulation, gradients) in β -adrenergic signaling shapes arrhythmia dynamics, a less stable model such as ToR-ORd which would require very careful selection of time step, mesh resolution, and conductivity to ensure that observed wavefront propagation patterns reflect physiological mechanisms rather than boundary effects or numerical block. Based on the results of Dong et al., such precautions would likely necessitate smaller time steps and higher spatial resolutions, substantially increasing the computational cost of simulations that are already more expensive. Furthermore, while Dong et al. did not explicitly explore resolution dependence, the average slab resolution in their study ($\sim 232 \pm 103 \mu$ m) was finer than the spatial resolution chosen for many MRI-based 3D geometric model reconstructions (Prakosa 2019), suggesting that nonphysiological behavior may become even more pronounced in such patient-specific simulations.

For these reasons, we believe TT2 remains a pragmatic and physiologically relevant choice for our goals, enabling us to explore the macroscopic consequences of heterogeneous β -adrenergic stimulation with appropriate spatial and temporal resolution while maintaining computational tractability.

As for the T-world model (2025), we note that it includes a built-in implementation of β -adrenergic signaling. Once the model code becomes publicly available, it would be valuable to conduct a benchmarking study similar to that of Dong et al., directly comparing the performance of T-world’s BARS implementation with that of our TT2-based framework.

5. Previous signaling models, such as (Heijman et al., 2011) do not include PKA-mediated modulation of IK1. Are there specific experimental references in canine myocytes that support PKA phosphorylation effects on IK1? For instance, (Reilly et al., 2020) reported a $\sim 20\%$ increase in IK1 in response to ISO in mice, but species differences should be considered when generalizing these effects. In the current model, a 150% increase in IK1 due to PKA phosphorylation appears quite high. Such a substantial enhancement can significantly reduce the resting membrane potential,

which in turn influences cellular excitability. I haven't come across experimental studies that examine the effects of ISO on resting membrane potential in cardiac cells. Could you please justify the magnitude of IK1 modulation implemented in your model and its physiological plausibility in the context of existing experimental evidence?

While in our old model the adjustment of IK1 had minimal effect on resting potential, it moderately affected the steepness of the end of phase 3. We recognize that its inclusion was not well justified physiologically and could potentially have important implications for reentrant dynamics especially in full organ simulations. In the revised model, we have removed this adjustment entirely, to maintain consistency with experimental reports of BARS effects and with other published models of β -adrenergic signaling. The manuscript has been updated accordingly.

6. Are the results presented in Figures 3 and 5 (panels A and B) obtained at a pacing rate of 1 Hz? If so, could you please comment on how these results compare at other pacing frequencies, especially at faster rates under sympathetic stimulation? It would be valuable to understand how the model predictions align with experimental observations across a range of pacing conditions.

Author response:

The results presented in Figures 3 and 5 are obtained at a pacing cycle length (PCL) of 1000ms (1Hz). Here we present the results for PCLs 400, 600, 800 ms. For each curve in the left and middle panels, the points are normalized to the model with ISO=0 for that PCL. As the model is paced faster, the Δ APD90 induced by ISO is amplified. Interestingly, while Δ CaT_{amplitude} induced by ISO decreases with shortening PCLs, a PCL of 400ms results in a Δ CaT_{amplitude} similar to that of PCL 800 ms at moderate to high ISO levels. The right panel shows [Ca²⁺]_i under the tested PCLs for selected ISO levels. For ISO values under 0.01 μ M, we observe a drop in [Ca²⁺]_i amplitude at 400ms, likely due to I_{CaL} reduction from accumulating subspace and intracellular calcium levels. However, at higher ISO levels with upregulated calcium cycling, there is less/no drop in [Ca²⁺]_i. The divergence in behavior results in a larger Δ CaT_{amplitude} observed in the middle panel.

7. Please provide details on the steady-state criteria used in the simulations. Specifically,

how many beats at each ISO concentration are required for the ionic concentrations and APD to stabilize? For reference, the Gong et al. (2020) model typically requires several hundred beats to reach steady state. Including this information would help clarify the simulation protocol and model behavior.

Author response:

To achieve steady state, cell-level simulations were paced for 1000 beats, allowing for the APD and CaT of the last 100 beats to vary less than 1%. We have included this information in the Methods section.

8. Reference 40 (Priori & Corr, 1990) demonstrated that in canine myocytes exposed to ISO for 10 minutes, intermediate ISO concentrations caused slight prolongation of APD, whereas higher concentrations led to APD shortening. Could you please clarify whether the proposed model is able to replicate this behavior? Including a comparison with this experimental observation would strengthen the model's validation.

Author response:

Our model does not reproduce the slight APD prolongation reported at intermediate ISO concentrations (e.g., 0.01 μM ISO). We believe this mismatch with experimentally observed behavior arises from the balance of phosphorylation among our BARS targets. From our fitted dose-response curves, a 0.01 μM ISO concentration results in 20% phosphorylation level of IKs, while calcium-handling targets such as I_{CaL} and I_{rel} remain less phosphorylated. Combined with the relatively strong baseline IKs in the ten Tusscher electrophysiology framework, this phosphorylation pattern likely shifts the net balance toward repolarizing currents, leading to APD shortening even at low ISO levels. The implementation of the Heijman BARS model in Gong et al. similarly does not appear to reproduce the intermediate APD prolongation. While these prior studies have examined the contributions of individual substrates to APD and CaT changes under maximal BARS, further analyses in both Gong's and our model should be performed at intermediate ISO levels.

9. At very low ISO concentrations (~0.01 μM), the model shows a slight decrease in conduction velocity compared to baseline in some cases. Intuitively, one would expect conduction to increase with any degree of sympathetic stimulation. Could you please clarify the reason for this observation? Additionally, it may be important to consider the effects of ISO on gap junctional coupling (diffusion coefficients) in the tissue simulations, alongside ion channel modulation. Previous studies (e.g., (Xia et al., 2009; Campbell et al., 2014) have demonstrated that β -adrenergic stimulation can influence gap junction conductance, which could significantly impact conduction velocity. Accounting for this could enhance the physiological relevance of the tissue-level model.

Author response:

In the revised model, we have adjusted our formulations so that conduction velocity (CV) restitution no longer exhibits a slight decrease at low ISO concentrations, which we agree was unexpected. We also agree that β -adrenergic signaling, as demonstrated in prior studies, can modulate gap junctional coupling as represented by tissue conductivity parameters. In our current tissue simulations, we chose not to incorporate conductivity changes between baseline and ISO conditions since our primary objective was to isolate and investigate the effects of ionic current modulation alone. However, we acknowledge that including β -adrenergic effects on conductivity would further improve the physiological relevance of tissue-scale simulations. In future applications of this framework to whole-organ simulations, we plan to incorporate spatially heterogeneous conductivity adjustments to represent differences in sympathetic innervation across regions.

10. The tissue-level simulations utilize very high ISO concentrations-1 μ M-distributed in different amounts across the tissue. Is such a saturating ISO level necessary for the model to generate fast and stable rotors? It would be helpful to understand how the model behaves at more physiological catecholamine concentrations and lower ISO levels in the range of 0.01 to 0.1 μ M. Could you please comment on the stability and dynamics of rotors under these conditions?

Author response:

In the previous version of our manuscript, we treated sympathetic stimulation as a binary phenomena thus we did not include the influence of different concentrations, which admittedly defeats the purpose of our proposed ionic model. Therefore, in our revised manuscript, we included data and analysis of changes to a rotor's angular speed and spatiotemporal stability as a function of ISO concentrations. We showed that a rotor's angular speed increases and spatiotemporal stability improves with increasing ISO concentrations. Moreover, our simulations suggest that increasing ISO concentrations combined with sharp spatial BARS gradients could influence when and how a rotor's wavetail could splinter into slower and unstable wavelets. Below is an excerpt from our revised manuscript discussing our findings on the influence of ISO concentrations on rotor dynamics. We also provide snippets from Fig. 7 of our revised manuscript.

Results - Tissue simulations - Rotors in ventricular substrates with BARS

[...] We show in Fig. 7A and Supplemental Video S1 that rotors hosted in uniformly BAR-stimulated substrates have faster angular speeds at higher ISO concentrations than plain substrates [...]. Although slower than those hosted on homogeneous substrates, we also show that the angular speed of hosted rotors increases with higher ISO concentrations, regardless of the degree of sympathetic remodelling [...]. A more detailed examination reveals that a rotor's angular speed gradually increases for ventricular substrates with spatial BARS densities greater than 15% [...]. Rotor angular speeds minimally change in substrates with greater spatial densities regardless of the ISO concentrations administered (Fig. 7B). [...]

Results - Tissue simulations - Effect of sympathetic spatial gradients on rotors

[...]. Specifically, wavelets could splinter from the primary rotor at the boundary where the APD and CV gradient are the greatest. For BAR stimulated substrates with 0.01 μ M ISO, wavelets begin to form only after a rotor has been sustained (Fig. 8A). In contrast, primary rotors are immediately splintered to wavelets in BAR stimulated substrates with ISO concentrations >0.1 μ M (Fig. 8B and C). [...]

11. Are the observed rotor dynamics and stability in the tissue simulations primarily driven by APD heterogeneity? For instance, if similar spatial APD heterogeneity were introduced by varying repolarizing conductances such as gKr or gKs (rather than through ISO clustering), how would the resulting rotor behavior compare? Given that ISO affects multiple targets-including Ca²⁺ handling, conduction velocity, APD, and restitution properties-it would be important to clarify whether the rotor dynamics observed are uniquely attributable to this multi-target modulation, as opposed to simpler models of APD heterogeneity (e.g., via gKr variation alone). A comparative analysis would help distinguish the specific role of sympathetic signaling in arrhythmia mechanisms.

Author response:

In our revised manuscript, we provided a comparison of rotor dynamics and stability between those hosted on BAR-stimulated substrates and those hosted on substrates with adjusted conductances of known APD modifiers (i.e., GKs and GKr). We simulated and analysed rotors sustained under 4 conditions (i.e., GKs \pm 50% and GKr \pm 50%) to show that the dynamics and stability of rotors hosted on BAR-stimulated substrates may be attributed to ISO's multi-target modulation as opposed to the effects of simple APD modifiers. Below, is an excerpt from our Discussions section detailing our observations. We also provide snippets from Fig. 7 of our revised manuscript.

Discussions

[...] We compared our findings with other known repolarization modifiers to evaluate whether they are uniquely attributable to BARS' multi-target activation or could be replicated with single target, such as varying the conductances of slowly (GKs) and rapid (GKr) activating delayed rectifier potassium currents. We increased and decreased the values of

GKs and GKr by 50%, respectively, and gradually increased their spatial densities as we did with adrenergically stimulated substrates. We showed in Fig. 7B that rotors hosted in sympathetic-remodelled are faster compared to those hosted in substrates with only GKs or GKr modified. While rotors in substrates with increasing spatial densities of GKs+50% elements have similar spatiotemporal organization as those hosted in adrenergically stimulated substrates, rotor trajectories are disrupted in substrates with other APD modifiers (i.e., GKs+50%, GKr-50%) as we showed in Fig. 7D. Interestingly, we observed that rotors self-terminate only in substrates with >15% spatial densities of elements with GKs-50%. Our observations suggest that rotor dynamics in BAR-stimulated substrates may be better linked to the multi-target modulation by ISO than to simple APD modification and heterogeneity. [...]

References used in reviewer response:

1. Campbell AS, Johnstone SR, Baillie GS & Smith G (2014). β -Adrenergic modulation of myocardial conduction velocity: Connexins vs. sodium current. *J Mol Cell Cardiol* 77, 147-154.
2. Doste R & Bueno-Orovio A (2021). Multiscale Modelling of β -Adrenergic Stimulation in Cardiac Electromechanical Function. *Mathematics* 9, 1785.
3. Gong JQX, Susilo ME, Sher A, Musante CJ & Sobie EA (2020). Quantitative analysis of variability in an integrated model of human ventricular electrophysiology and β -adrenergic signaling. *J Mol Cell Cardiol* 143, 96-106.
4. Heijman J, Volders PGA, Westra RL & Rudy Y (2011). Local control of β -adrenergic stimulation: Effects on ventricular myocyte electrophysiology and Ca^{2+} -transient. *J Mol Cell Cardiol* 50, 863-871.
5. Khalilimeybodi, A., Daneshmehr, A., & Sharif-Kashani, B. (2018). Investigating β -adrenergic-induced cardiac hypertrophy through computational approach: classical and non-classical pathways. *The journal of physiological sciences : JPS*, 68(4), 503–520. <https://doi.org/10.1007/s12576-017-0557-5>
6. Koncz, I., Verkerk, A. O., Nicastro, M., Wilders, R., Árpádfy-Lovas, T., Magyar, T., Tóth, N., Nagy, N., Madrid, M., Lin, Z., & Efimov, I. R. (2022). Acetylcholine Reduces IKr and Prolongs Action Potentials in Human Ventricular Cardiomyocytes. *Biomedicines*, 10(2), 244. <https://doi.org/10.3390/biomedicines10020244>
7. Morotti S, Liu C, Hegyi B, Ni H, Fogli Iseppe A, Wang L, Pritoni M, Ripplinger CM, Bers DM, Edwards AG & Grandi E (2021). Quantitative cross-species

translators of cardiac myocyte electrophysiology: Model training, experimental validation, and applications. *Sci Adv* 7, eabg0927.

8. Prakosa, A., Arevalo, H. J., Deng, D., Boyle, P. M., Nikolov, P. P., Ashikaga, H., Blauer, J. J. E., Ghafoori, E., Park, C. J., Blake, R. C., 3rd, Han, F. T., MacLeod, R. S., Halperin, H. R., Callans, D. J., Ranjan, R., Chrispin, J., Nazarian, S., & Trayanova, N. A. (2018). Personalized virtual-heart technology for guiding the ablation of infarct-related ventricular tachycardia. *Nature biomedical engineering*, 2(10), 732–740. <https://doi.org/10.1038/s41551-018-0282-2>
9. Priori SG & Corr PB (1990). Mechanisms underlying early and delayed afterdepolarizations induced by catecholamines. *Am J Physiol* 258, H1796-1805.
10. Reilly L, Alvarado FJ, Lang D, Abozeid S, Van Ert H, Spellman C, Warden J, Makielski JC, Glukhov AV & Eckhardt LL (2020). Genetic Loss of IK1 Causes Adrenergic-induced Phase 3 Early Afterdepolarizations and Polymorphic and Bi-directional Ventricular Tachycardia. *Circ Arrhythm Electrophysiol* 13, e008638.
11. Saucerman JJ, Brunton LL, Michailova AP & McCulloch AD (2003). Modeling β -Adrenergic Control of Cardiac Myocyte Contractility in Silico*. *J Biol Chem* 278, 47997-48003.
12. Tomek J, Zhou X, Martinez-Navarro H, Holmes M, Bury T, Berg LA, Tomkova M, Jo E, Nagy N, Bertrand A, Bueno-Orovio A, Colman M, Rodriguez B, Bers D & Heijman J (2025). T-World: A highly general computational model of a human ventricular myocyte. 2025.03.24.645031. Available at: <https://www.biorxiv.org/content/10.1101/2025.03.24.645031v2> [Accessed June 8, 2025].
13. Volders, P.G., Stengl, M., van Opstal, J.M., Gerlach, U., Spätjens, R.L., Beekman, J.D., Sipido, K.R., Vos, M.A., 2003. Probing the: contribution of IKs to canine ventricular repolarization. *Circulation* 107, 2753–2760
14. Xia Y, Gong K, Xu M, Zhang Y, Guo J, Song Y & Zhang P (2009). Regulation of gap-junction protein connexin 43 by β -adrenergic receptor stimulation in rat cardiomyocytes. *Acta Pharmacol Sin* 30, 928-934.

END OF COMMENTS

Dear Dr Magtibay,

Re: JP-RP-2025-289340R1 "A model of beta-adrenergic stimulation in human ventricular cells for tissue-scale simulations of sympathetically-modulated tachycardias" by Kelly Zhang, Karl Magtibay, Natalia A Trayanova, and Edward Vigmond

Thank you for submitting your manuscript to The Journal of Physiology. It has been assessed by a Reviewing Editor and by 2 expert referees and we are pleased to tell you that it is acceptable for publication following satisfactory revision.

REVISION CHECKLIST:

We look forward to receiving your revised submission.

Yours sincerely,

Bjorn Knollmann
Senior Editor
The Journal of Physiology

EDITOR COMMENTS

Reviewing Editor:

Thank you for addressing the reviewers' comments. Additional minor revisions are needed for clarification.

Senior Editor:

I concur with the reviewing editor

REFEREE COMMENTS

Referee #1:

Most of my concerns have been addressed but a few issues remain:

1) Minor point 2 from the original submission:

The revised text is still not clear (lines 169-171):

"We used a cell element to tune the intracellular parameters of our novel ionic model to reach a steady state, stimulated with a $60 \mu\text{A}/\text{cm}^2$, 0.5 ms transmembrane current, with a 600 ms basic CL over 60 s. Then, we used our ionic model's steady-state parameters"

The first part sounds like you are doing additional tuning, when - I think - this simply refers to the model development already described. In the follow-up sentence, it is not clear what is meant by the "model's steady-state parameters" or how they were used - was the 2D model run using the 0D steady-state values of the state variables as the initial condition?

2) Minor point 6 from the original submission:

I don't see where in the manuscript the parameters for the dose-response curves are given. They should appear somewhere.

Additional issues/questions:

- 1) The reference Plank et al contains some undefined asterisks
- 2) Gating variables are generally not capitalized (lines 117-118).
- 3) Please check if the denominator in line 211 should be t -tau instead of tau- t .
- 4) Please clarify what is meant by subject in line 229. Substrate?
- 5) The abstract states that the APD is reduced by 32% with ISO. The change in Figure 4E is smaller than that. The condition under which the change is 32% should be given.
- 6) Lines 275-276 state that "since the transient duration is also shortened under BARS, there is a decrease in APD". How do the authors know that the APD reduction is due to the shortened CaT duration?
- 7) Lines 276-277 then state "Between 0.01 μ M and 0.1 μ M ISO, the reduced CaT growth along with the increased fraction of phosphorylated channels of repolarizing currents causes significant APD shortening." What is the reduced CaT growth? The CaT amplitude increases a lot over this [ISO] range...
- 8) Fig 6: I don't see any substantial differences in phase singularity trajectory between the three panels. The text (lines 298-299) compares BARS to no BARS, but no BARS is not included in the figure.
- 9) The reader has to deduce what is meant by "Any % BARS" and "Full" in Figure 7. Please define these. I don't see what is meant by "A more detailed examination reveals that a rotor's angular speed gradually increases for ventricular substrates with spatial BARS densities greater than 15% ($> 4.2 \{plus\ minus\} 0.04$ Hz)." The angular speed also increases for spatial density $< 15\%$ (Fig 7B).
- 10) Are Figures 7E-G for one specific ISO concentration? Please clarify. Similarly: are H and I for one specific spatial density difference?
- 11) I was not able to play to supplemental videos. Please make sure they are in a format that is friendly to a variety of systems.

Referee #2:

The authors have addressed the queries and concerns thoroughly.

It would be important to note in the discussion that the current signalling model is simplified and does not capture additional regulatory mechanisms such as changes in cAMP levels, PDE concentrations, or other downstream alterations in phosphorylation, as the model primarily fits dose-response ISO curves.

END OF COMMENTS

Title: A model of beta-adrenergic stimulation in human ventricular cells for tissue-scale simulations of sympathetically-modulated tachycardias

Authors: Kelly Zhang*, Karl Magtibay*, Natalia Trayanova, and Edward Vigmond (*shared first authors)

Manuscript ID: JP-RP-2025-289340R1

Response to Reviewers

As in our first revision, the following sections contain our responses to each reviewer's comment. Original reviewer comments are **bolded in blue**, and our responses are in black. We wrote in *italics* the appropriate sections where the changes in our revised manuscript could be found. The [...] notation indicates continuation of paragraphs or sentences that may be unrelated to the reviewer's queries.

EDITOR COMMENTS

Reviewing Editor:

Thank you for addressing the reviewers' comments. Additional minor revisions are needed for clarification.

Senior Editor:

I concur with the reviewing editor

Author response:

We thank the reviewers and the editors for their positive feedback. Below, we address remaining issues brought up by each referee and cite the revisions in our revised manuscript, where applicable and relevant to our study.

REFEREE COMMENTS

Referee #1:

Most of my concerns have been addressed but a few issues remain:

Author response:

We thank the reviewer for their positive response. Below, we address the remaining issues from our revised manuscript.

1) Minor point 2 from the original submission:

The revised text is still not clear (lines 169-171):

"We used a cell element to tune the intracellular parameters of our novel ionic model to reach a steady state, stimulated with a 60 μ A/cm², 0.5 ms transmembrane current, with a 600 ms basic CL over 60 s. Then, we used our ionic model's steady-state parameters"

The first part sounds like you are doing additional tuning, when - I think - this simply refers to the model development already described. In the follow-up sentence, it is not clear what is meant by the "model's steady-state parameters" or how they were used - was the 2D model run using the 0D steady-state values of the state variables as the initial condition?

Author response:

We revised lines 170-172 to clarify that we paced our ionic model for its state variables to reach stable values rather than using the verb “tuned”, and that we used our model’s state variable values at rest as initial conditions for our 2D simulations. Below is an excerpt from our revised manuscript.

Methods, Applications to tissue-level simulations, Two-dimensional substrates, BARS spatial density, and rotor generation

We paced a cell using a 60 uA/cm² 0.5 ms transmembrane current at 600 ms basic CL over 60 s to reach a steady state. Then, we used the cell's state variable values at rest as initial conditions [...]

2) Minor point 6 from the original submission:

I don't see where in the manuscript the parameters for the dose-response curves are given. They should appear somewhere.

Author response:

We reworked Figure 1 and its caption to include the dose-response curves and their equations.

Figure 1, figure caption

Figure 1: A. Model of β -adrenergic receptor signaling (BARS) pathway in the cardiomyocyte. The model captures the downstream electrophysiological effects of the BARS pathway (grey box) on the intracellular dynamics of ventricular cardiomyocytes. Modulated ionic currents and fluxes are outlined in orange. The magnified inset depicts the calculation of total currents from the fractional contribution of non-phosphorylated and phosphorylated channel populations. B. The BARS pathway is represented by dose-response curves relating ISO to the fractional contribution of phosphorylated channel populations.

Additional issues/questions:

1) The reference Plank et al contains some undefined asterisks

Author response:

The asterisks were artifacts of the paper’s citation format, indicating the corresponding authors. We removed these in our revised manuscript.

2) Gating variables are generally not capitalized (lines 117-118).

Author response:

We revised the gating variables to lowercase.

3) Please check if the denominator in line 211 should be t-tau instead of tau-t.

Author response:

We corrected the Hilbert transform (t-tau in the denominator) expression as shown by Kuklik P, et al (2014).

4) Please clarify what is meant by subject in line 229. Substrate?

Author response:

We corrected the term “subject” to “substrate”.

5) The abstract states that the APD is reduced by 32% with ISO. The change in Figure 4E is smaller than that. The condition under which the change is 32% should be given.

Author response:

This value was mistakenly retained from our previous draft. The percent change in APD90 under 1 uM ISO in our revised model should be a 16% reduction (51 ms). We updated our abstract to be consistent with Figures 4E and 5A.

6) Lines 275-276 state that "since the transient duration is also shortened under BARS, there is a decrease in APD". How do the authors know that the APD reduction is due to the shortened CaT duration?

Author response:

We revised lines 275-276 to state that “*the combined effect of a shortened CaT duration and an increase in repolarizing IKs resulted in the APD decrease*”.

7) Lines 276-277 then state "Between 0.01 μ M and 0.1 μ M ISO, the reduced CaT growth along with the increased fraction of phosphorylated channels of repolarizing currents causes significant APD shortening." What is the reduced CaT growth? The CaT amplitude increases a lot over this [ISO] range...

Author response:

The “reduced growth” between 0.01 M and 0.1 M ISO is incorrect. We revised lines 276-277 of our manuscript to state that “*the increased CaT growth is further countered by the increased phosphorylated channels of repolarizing currents, resulting in significant APD shortening.*”

8) Fig 6: I don't see any substantial differences in phase singularity trajectory between the three panels. The text (lines 298-299) compares BARS to no BARS, but no BARS is not included in the figure.

Author response:

We revised Figure 6 to include a substrate without any BARS and show the isometric view of a rotor's path, and enhanced the differences of phase singularity trajectories over different spatial BARS densities.

9) The reader has to deduce what is meant by "Any % BARS" and "Full" in Figure 7. Please define these. I don't see what is meant by "A more detailed examination reveals that a rotor's angular speed gradually increases for ventricular substrates with spatial BARS densities greater than 15% ($> 4.2 \pm 0.04$ Hz)." The angular speed also increases for spatial density $< 15\%$ (Fig 7B).

Author response:

We revised lines 311-315 to indicate that rotor angular speed increases for ventricular substrates with spatial BARS densities less than 15%, so that the following sentence would be consistent with our observations. Below is an excerpt from our revised manuscript.

Results, Tissue simulations, Rotors in ventricular substrates with BARS

[...] A more detailed examination reveals that a rotor's angular speed gradually increases for ventricular substrates with spatial BARS densities $\leq 15\%$ (from 3.9 ± 0.04 Hz to 4.2 ± 0.04 Hz). Rotor angular speeds minimally change in substrates with spatial BARS densities $> 15\%$ regardless of ISO concentrations administered (Fig. 7B).

10) Are Figures 7E-G for one specific ISO concentration? Please clarify. Similarly: are H and I for one specific spatial density difference?

Author response:

We clarified in our revised manuscript and their corresponding legends that Figures 7E-G represent measurements for all ISO concentrations and that Figures 7H-I represent measurements for all spatial density gradients.

11) I was not able to play to supplemental videos. Please make sure they are in a format that is friendly to a variety of systems.

Author response:

We re-rendered and verified the functionality of our supplemental videos to be friendly to a variety of video players.

Referee #2:

The authors have addressed the queries and concerns thoroughly.

Author response:

We thank the reviewer for their positive response. Below, we address the remaining issues from our revised manuscript.

It would be important to note in the discussion that the current signalling model is simplified and does not capture additional regulatory mechanisms such as changes in cAMP levels, PDE concentrations, or other downstream alterations in phosphorylation, as the model primarily fits dose-response ISO curves.

Author response:

The *Limitations* section has been edited to highlight the simplifications made to our model.

Limitations

[...] Our approach necessarily simplifies several aspects, most notably in the representation of the BARS pathway as a set of dose-response curves. We do not incorporate receptor adrenergic receptor isoforms (i.e., β_1 , β_2 , and α_1), their localization in functional intracellular domains (e.g. subspace, caveolae, cytosol) of the myocyte, or the impact of changes in downstream signalling proteins. We omit other domain-specific signalling cascades targeting the same PKA substrates, such as the ryanodine receptor, that are regulated by calmodulin kinase. Consequently, the model does not characterize the synergistic effects of these two pathways on the CaT. Additionally, the model does not account for feedback mechanisms that lead to BARS desensitization. [...]

END OF COMMENTS

Dear Dr Magtibay,

Re: JP-RP-2025-289340R2 "A model of beta-adrenergic stimulation in human ventricular cells for tissue-scale simulations of sympathetically-modulated tachycardias" by Kelly Zhang, Karl Magtibay, Natalia A Trayanova, and Edward Vigmond

Thank you for submitting your manuscript to The Journal of Physiology. It has been assessed by a Reviewing Editor and by 2 expert referees and we are pleased to tell you that it is acceptable for publication following satisfactory revision.

REVISION CHECKLIST:

We look forward to receiving your revised submission.

Yours sincerely,

Bjorn Knollmann
Senior Editor
The Journal of Physiology

REQUIRED ITEMS

1) - Please ensure that all figures and tables have a title and legend, and that they have been cited within the main article text.

EDITOR COMMENTS

Reviewing Editor:

Thank you for revising the manuscript. There is one remaining suggestion that the authors may wish to consider.

Senior Editor:

The MS is acceptable pending a minor clarification as suggested by reviewer 2.

REFEREE COMMENTS

Referee #1:

The authors have addressed my concerns and made appropriate revisions to the manuscript.

Referee #2:

The authors have addressed the queries and concerns from Reviewer 1. It would also be important to include an additional limitation of the model, since it fits only steady state dose response curves; it cannot capture transient ISO effects. This is particularly important, since there can be arrhythmias due to a sudden adrenergic surge (example Ref Xie et al 2014 <https://doi.org/10.1093/europace/eut412>), please include this as a possibility or future goal.

END OF COMMENTS

Title: A model of beta-adrenergic stimulation in human ventricular cells for tissue-scale simulations of sympathetically-modulated tachycardias

Authors: Kelly Zhang*, Karl Magtibay*, Natalia Trayanova, and Edward Vigmond (*shared first authors)

Manuscript ID: JP-RP-2025-289340R2

Response to Reviewers

As in our first and second revisions, the following sections contain our responses to each comment the reviewers provided. Original reviewer comments are **bolded** in blue, and our responses are in black. We wrote in *italics* the appropriate sections where the changes in our revised manuscript could be found. The [...] notation indicates continuation of paragraphs or sentences that may be unrelated to the reviewer's queries.

EDITOR COMMENTS

Reviewing Editor:

Thank you for revising the manuscript. There is one remaining suggestion that the authors may wish to consider

Senior Editor:

The MS is acceptable pending a minor clarification as suggested by reviewer 2.

Author response:

Below, we address reviewer 2's suggestion and cite the revisions in our revised manuscript, where applicable and relevant to our study. In addition, we corrected existing grammatical and syntactic errors to prepare our manuscript for eventual publication.

REFEREE COMMENTS

Referee #1:

The authors have addressed my concerns and made appropriate revisions to the manuscript.

Author response:

We thank the reviewer for all their critical feedback throughout this process, which has further improved the quality of our work.

Referee #2:

The authors have addressed the queries and concerns thoroughly.

The authors have addressed the queries and concerns from Reviewer 1. It would also be important to include an additional limitation of the model, since it fits only steady state dose response curves; it cannot capture transient ISO effects. This is particularly important, since there can be arrhythmias due to a sudden adrenergic surge (example

Ref Xie et al 2014 <https://doi.org/10.1093/europace/eut412>), please include this as a possibility or future goal.

Author response:

We have added this limitation to our manuscript.

Limitations

[...] *Our approach necessarily simplifies several aspects, most notably in the representation of the BARS pathway as a set of steady-state dose-response curves, which are unable to capture the transient effects of ISO. Adrenergic surges, which have been shown to promote arrhythmogenesis by disrupting the balance of inward and outward currents (Xie et al. 2014, Dajani et al. 2023), cannot be directly studied by the current model. Future extensions of the framework could include the time course of phosphorylation for the various target channels.*
[...]

END OF COMMENTS

Dear Dr Magtibay,

Re: JP-RP-2026-289340R3 "A model of beta-adrenergic stimulation in human ventricular cells for tissue-scale simulations of sympathetically-modulated tachycardias" by Kelly Zhang, Karl Magtibay, Natalia A Trayanova, and Edward Vigmond

We are pleased to tell you that your paper has been accepted for publication in The Journal of Physiology.

Yours sincerely,

Bjorn Knollmann
Senior Editor
The Journal of Physiology

IMPORTANT POINTS TO NOTE FOLLOWING ACCEPTANCE OF YOUR PAPER:

- **IMPORTANT NOTICE ABOUT OPEN ACCESS:** To assist authors whose funding agencies mandate immediate public access to published research findings, The Journal of Physiology allows authors to pay an Open Access (OA) fee to have their papers made freely available immediately on publication.

- You can help your research get the attention it deserves! Check out Wiley's free Promotion Guide for best-practice recommendations for promoting your work at: www.wileyauthors.com/eeo/guide. You can learn more about Wiley Editing Services which offers professional video, design, and writing services to create shareable video abstracts, infographics, conference posters, lay summaries, and research news stories for your research at: www.wileyauthors.com/eeo/promotion.

- If you would like to receive our 'Research Roundup', a monthly newsletter highlighting the cutting-edge research published in The Physiological Society's family of journals (The Journal of Physiology, Experimental Physiology, Physiological Reports, The Journal of Nutritional Physiology and The Journal of Precision Medicine: Health and Disease), please click this link, fill in your name and email address and select 'Research Roundup': <https://www.physoc.org/journals-and-media/membernews>

EDITOR COMMENTS

Reviewing Editor:

Thank you for further revising the manuscript.

Senior Editor:

Thank you for your excellent contribution to the Journal!